# The involvement of attentional biases in endogenous pain inhibition and autonomic reactivity

Einav Gozansky[1,2,3*◉], Irit Weissman-Fogel[2,3,4◉], Hadas Okon-Singer[1,2,3,5◉]

1 Department of Psychology, School of Psychological Sciences, University of Haifa, Haifa, Israel, 2 The Integrated Brain and Behavior Research Center (IBBR), University of Haifa, Haifa, Israel, 3 The University of Haifa's AI Research Center (HiAI), Haifa, Israel, 4 Physical Therapy Department, Faculty of Social Welfare and Health Sciences, University of Haifa, Haifa, Israel, 5 Max Planck Institute for Human Cognitive and Brain Sciences, Leipzig, Germany

◉ These authors contributed equally to this work.
* einavgoz@gmail.com

## Abstract

### Background

Attention is a key factor in shaping pain perception and modulation, yet its role in explaining individual differences in endogenous pain inhibition—commonly assessed in humans using conditioned pain modulation (CPM)—remains poorly understood. While studies show that attentional focus during dual pain stimulation can influence CPM efficacy, the contribution of attentional biases has not been examined. In addition, although psychological traits and autonomic nervous system activity have both been linked to CPM, their relative contributions to individual variability remain unclear. The present study examined whether attentional biases predict individual differences in pain sensitivity and endogenous pain modulation, and whether psychological, emotional, and autonomic factors are associated with these outcomes.

### Methods

Eighty-six healthy women completed psychological questionnaires assessing anxiety, depression, and fear of pain. They also performed two modified attentional bias tasks: a novel perceptual load task and a dot-probe task. Quantitative sensory testing included pain ratings of suprathreshold tonic heat pain alone and under conditioning (CPM paradigm). Heart rate (HR) was measured at baseline, during, and after both test-stimulus alone and under conditioning.

### Results

Greater interference from pain-related cues under high perceptual load was linked to increased pain sensitivity, whereas greater attentional avoidance in the dot-probe

**Data availability statement:** The data is fully available in our data archive on the Open Science Framework (OSF) website: https://osf.io/3bwy7/overview.

**Funding:** This work was supported by the AI Research Center (HiAI) at the University of Haifa and awarded to all authors, as well as by the Azrieli Fellows Graduate Studies Program, which was granted to EG. The sponsors played no role in the study design, data collection and analysis, decision to publish, or preparation of the manuscript.

**Competing interests:** The authors have declared that no competing interests exist.

task was associated with higher fear of pain. In contrast, none of the examined factors predicted CPM magnitude. HR increased during the test-stimulus under conditioning and remained elevated in recovery. Exploratory analyses further showed that higher emotional distress was related to blunted HR during test-stimulus under conditioning, and that both perceptual load interference and attentional avoidance predicted elevated HR during recovery.

## Conclusions

Attentional biases to pain predicted pain sensitivity beyond psychological and autonomic influences. Although unrelated to CPM, they were associated with HR dynamics, suggesting a role in autonomic regulation during pain and recovery.

---

## 1. Introduction

Attention to pain plays a crucial evolutionary role by interrupting ongoing behavior and prioritizing responses to potential threats [1,2]. Experimental evidence consistently demonstrates that attentional focus modulates pain perception: directing attention toward pain tends to amplify the experience, whereas distraction can attenuate it [3–5]. Attention allocation to pain is further shaped by task demands and individual goals, influencing the salience of pain-related stimuli, pain processing, and behavioral responses [1,2,6–8]. For instance, individuals who habitually focus on pain tend to show greater task disruption during pain, whereas those who tend to divert attention away exhibit better task performance under the same conditions [9].

Beyond its role in shaping the subjective experience of pain, attention influences pain processing and modulation [2,10]. One widely used experimental paradigm to assess descending pain inhibition efficacy is the conditioned pain modulation (CPM) paradigm, in which a conditioning pain stimulus inhibits the perception of a painful test stimulus delivered at a remote body site [11,12]. CPM responses vary considerably between individuals, likely due to multiple interacting factors affecting the descending pain modulatory system function, including cognitive, emotional, and physiological processes [12–15]. Emerging evidence suggests that attention may contribute to this variability, as focusing on the conditioning stimulus enhances CPM-induced analgesia, whereas focusing on the test stimulus reduces it, resulting in enhanced pain feeling [16]. However, findings on attentional distraction remain inconsistent: while some studies report that distraction enhances CPM efficacy [17,18], others find no such effect [19,20].

Although attentional focus has been shown to influence CPM, the contribution of individual attentional tendencies to CPM responses remains largely unexplored [16]. Recent findings suggest their relevance, indicating that individuals with a greater intrinsic ability to disengage from pain exhibit more efficient CPM [21]. Such tendencies are commonly assessed through attentional bias paradigms, which capture the inclination to prioritize threat-related cues, including pain-related information [22,23]. Attentional biases have been linked to greater pain sensitivity [24–26], and can be

experimentally modified to alter pain perception [27,28]. However, their contribution to individual differences in endogenous pain modulation was not explored.

Besides attention, CPM magnitude is also shaped by physical properties of the test, including the modality of the test stimulus [29], its intensity [30], and the perceived subjective pain experience [13]. Importantly, recent work highlights the role of individual differences, showing that they account for roughly one-third of the variance in CPM, whereas demographic and stimulus-related factors together explain only about 10% [14]. This is further demonstrated by works showing that variance in CPM magnitude is attributed to psychological and state emotional factors, such as anxiety [31,32], depression [31], and pain catastrophizing [33,34]. Given that attentional biases may also be influenced by psychological traits and emotional states [35–38], it is important to disentangle their relative contributions to CPM modulation by controlling for these psychological factors [39].

In addition, autonomic nervous system (ANS) activity is increasingly recognized as a key contributor to the subjective experience of pain [40,41] and pain inhibition [42–44]. Specifically, initial evidence suggests that the ANS may be involved in shaping CPM efficacy. A few studies have linked resting vagal tone, measured via heart rate variability (HRV) or blood pressure, to later CPM magnitude, with higher vagal tone at rest predicting a larger CPM effect [32,45,46]. Other work highlights the role of sympathetic activation, showing that greater sympathetic reactivity during pain alone is associated with enhanced CPM efficacy [47,48]. At the same time, cardiovascular autonomic dynamics during CPM have been investigated in only two studies to date [49,50]. Both reported group-level decreases in vagal tone and increases in sympathetic activation during the CPM procedure; however, only one study examined whether such changes were associated with CPM magnitude, and found no significant correlation [50].

With regard to attention involvement in ANS modulation, evidence consistently shows that attention can influence autonomic reactivity through shared neural mechanisms [51,52]. For example, attentional biases toward threat-related cues have been linked to heart rate (HR) acceleration and increased fear responses when individuals are exposed to their feared stimuli [53]. However, it remains unclear how attentional biases influence ANS responses during pain modulation, highlighting a critical gap in understanding how cognitive and physiological mechanisms jointly shape pain inhibition.

This study aimed to determine whether attentional bias accounts for individual differences in pain sensitivity and endogenous pain modulation, and to what extent psychological, emotional, and autonomic factors offer additional explanatory power. We focused particularly on CPM as an index of endogenous pain inhibition and HR dynamics as a marker of autonomic activity. We primarily hypothesized that individuals with stronger attentional biases toward pain would exhibit reduced endogenous pain inhibition. In addition, we expected that higher levels of pain-related distress, heightened autonomic reactivity during CPM, would contribute secondary, complementary effects, further explaining individual variability in endogenous pain modulation.

## 2. Methods

### 2.1. Participants

Participants were recruited through the university's research SONA participation system (an online system used to manage and screen student research participation), between October 15, 2022, and December 30, 2023. Inclusion criteria were: (i) age 18–40, (ii) no cardiovascular, metabolic, or neurological conditions or ADHD diagnosis, (iii) no reports of chronic pain in the past three months, (iv) not pregnant, and (v) no constant use of analgesics or psychiatric medication in the past three months. Participants were asked to avoid caffeine for 3 hours and pain medication for 24 hours before testing. Eligibility was confirmed both before and upon arrival at the lab, using a structured self-report medical history questionnaire.

A priori power analysis was conducted using G*Power version 3.1.9.7 [54] to determine the required sample size. The analysis indicated that a sample size of N = 84 was needed to achieve 80% power for detecting small-to-medium effects in

two multiple linear regression analyses with 8 predictors (i.e., Pain-60 temperature, attention bias in the low and high load conditions of the Perceptual load task, attention bias in the Dot-Probe task, Fear of Pain Questionnaire–9 (FPQ-9), Pain Catastrophizing Scale (PCS), Depression Anxiety Stress Scales–21 (DASS-21), and HR reactivity), assuming a significance level of α = .05. To account for potential dropout, the target sample was increased by 10% to 93 participants.

Ninety-four healthy females participated in the study in exchange for course credit or financial compensation, as part of a larger study examining the effect of cognitive-emotional trainings on pain perception. Eight participants were excluded due to technical issues or low adherence, resulting in a final sample of 86 participants (Mean age 23.21 ± 3 years old; age range 18–33).

## 2.2. Experimental procedure

The study was approved by the University of Haifa Ethics Committee (approval number 438/21), and all participants provided written informed consent prior to participation. Upon arrival, participants were fitted with electrocardiography (ECG) electrodes and instructed to sit quietly for five minutes while their resting heart rate was continuously recorded (*rest HR*).

Next, participants were seated approximately 60 cm from a computer screen and completed the baseline assessment, including demographics (e.g., age, Body mass index (BMI), ethnicity), psychological traits, and emotional state (PCS, FPQ-9, and DASS-21; see Fig 1). BMI was recorded because it influences heart rate [55], allowing us to evaluate potential confounding effects on autonomic measures. Participants then completed two attentional bias tasks (Perceptual Load and Dot-probe), with instructions provided both on-screen and verbally. At the beginning of each task, practice trials were used to ensure task comprehension and were repeated once if performance accuracy was low. Ethnicity (Arab vs. Jewish) was recorded to describe the sample and to assess whether mother tongue influenced performance on the Dot-Probe task, which contained Hebrew words.

Subsequently, participants completed a Quantitative Sensory Testing (QST) assessment, which began with a familiarization phase. This was followed by determining the suprathreshold pain stimulus intensity, which was tailored to the individual to induce a pain rating of 60 on a 0–100 numerical pain scale (NPS; pain-60). It was then applied to the forearm for 60 seconds, both alone and during the CPM paradigm. Pain ratings were reported verbally every 10 seconds using the NPS. ECG was recorded during the *baseline* (1 minute before the THP), *reactivity* (during the THP), and *recovery* (1 minute after the THP) phases for both the application of THP alone and during the CPM paradigm (Fig 1). Participants were then debriefed and compensated.

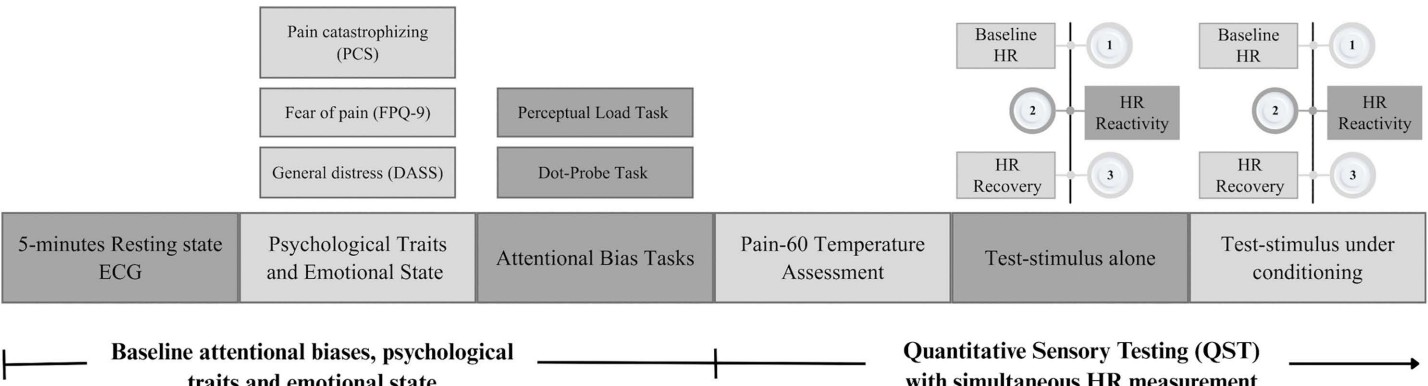

**Fig 1. Experimental timeline procedure.** The procedure included two main phases: First, participants completed the baseline assessments, including psychological traits, emotional state, and two attentional bias tasks. Second, quantitative sensory testing was conducted alongside simultaneous ECG recording.

## 2.3. Measures

### 2.3.1. Attentional bias tasks.
For ease of reading, attentional bias task details are presented here briefly. All procedural details, including stimulus timing, word presentation locations, counterbalancing scheme, word-set construction, probe identities, practice trial structure, and the complete list of stimuli, are provided in Supplementary Materials (S1 Text and S1 and S2 Tables) to ensure full transparency and reproducibility.

**2.3.1.a. Modified Perceptual Load task.** Attentional capture by pain-related information under differing cognitive demands was assessed using a modified version of the Perceptual Load paradigm [56], adapted from our previous work [57,58]. In this task, participants identified a target letter (X or N) presented under either low or high perceptual load, operationalized by the number of non-target letters surrounding the target. High-load trials required greater attentional resources, thereby reducing available capacity for processing irrelevant information.

On each trial, a pain-related or neutral image briefly appeared at fixation while participants performed the letter-identification task. Participants were instructed to ignore the images and respond as quickly and accurately as possible. Attentional interference was quantified as a pain-related interference index, calculated as the difference in reaction times (RTs) between pain-image and neutral-image trials, separately for low- and high-load conditions (see Fig 2).

**2.3.1.b. Dot-Probe task.** Attentional orienting toward pain-related information was further assessed using a modified Dot-Probe task commonly applied in pain research [59,60]. The task measures attention bias by simultaneously displaying one pain word and one neutral word on the screen, after which one of them is replaced by a letter probe. In this task, although participants were not required to respond to pain or neutral stimuli directly, attentional capture is inferred from RT to the probe appearing in a location previously occupied by either a pain or a neutral word.

Participants were asked to identify the probe letter as quickly and accurately as possible, and an attention bias score was computed by subtracting the mean reaction time of pain-congruent trials (probe replacing the pain word) from pain-incongruent trials (probe replacing the neutral word Fig 3).

### 2.3.2. Quantitative sensory testing (QST).

**2.3.2.a. Suprathreshold heat pain sensitivity assessment.** A 30 × 30 cm Peltier thermode (Thermal Sensory Analyzer-2001, Medoc, Israel) was applied to the volar surface of the dominant forearm, to determine the heat temperature corresponding to pain-60 (i.e., the temperature participants rate as 60 on a 0–100 numerical pain scale (NPS), ranging from '0' = no pain, '100' = worst imaginable pain). Stimulus intensity was individually calibrated such that all participants

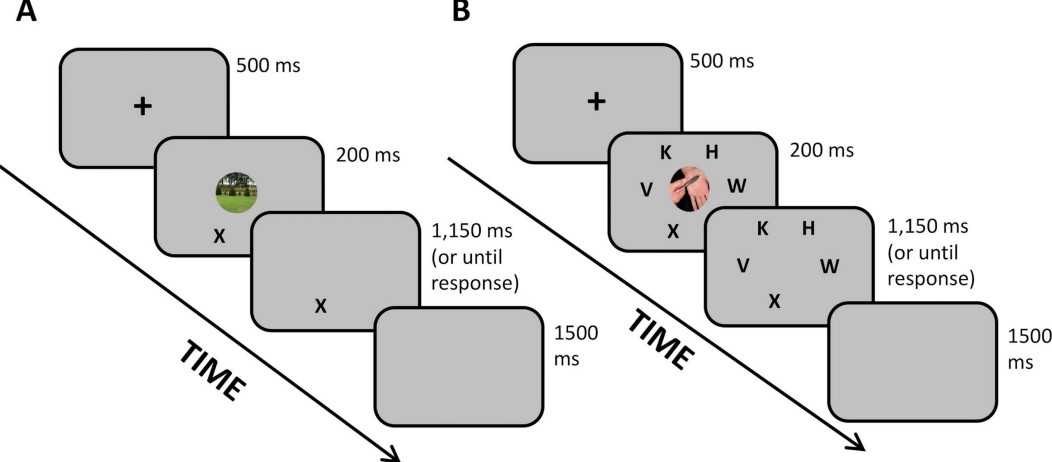

**Fig 2. Examples of trials in both loads of the Perceptual Load task.** *A.* Low load condition with a neutral picture trial; One target letter with a neutral distracting picture appearing at the center. *B.* High load condition with a pain picture trial; Four distracting letters with one target letter (the letter X) and a pain-related distracting picture appearing at the center.

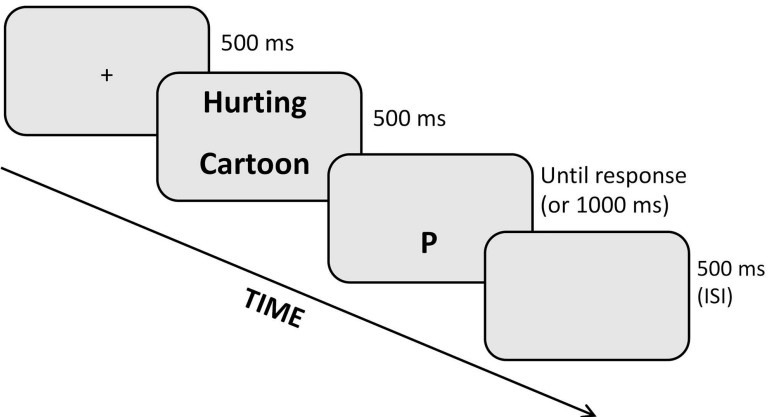

**Fig 3. Example of a trial in the modified Dot probe task.** In this example, the probe appears in the location of the neutral word (i.e., pain-incongruent trial). ISI = Inter-Stimulus Interval.

experienced the same subjective level of pain, as the focus of the study was on pain perception rather than nociceptive input. The pain-60 calibration procedure employed here is consistent with the commonly used protocol in the literature [61]. In general, participants were exposed to a series of hot stimuli of 8-second duration. Participants were first familiarized with the NPS by rating three initial heat stimuli (43 °C, 45 °C, and 47 °C; ramp rate: 2 °C/s from a 32 °C baseline), with a 20-seconds inter-stimulus interval. After each stimulus, participants were asked to report their level of pain on the verbal NPS. Based on these ratings, we identified an initial temperature with the NPS score closest to 60, with which we continued our pain-60 determination procedure.

The pain-60 determination then continued with a series of blocks containing three heat stimuli: the closest temperature to the temperature rated as 60 at the introduction phase, one degree lower, and one degree higher (presented in a random order). After each trial, the thermode was repositioned to prevent sensitization. At the end of each rating block, if none of the three temperatures elicited a rating near 60, another block containing three ratings was conducted, with the heat stimuli temperatures adjusted upward or downward in 1 °C increments, again including the adjacent temperatures above and below the new center point.

Importantly, this procedure was iterative: blocks were repeatedly administered and adjusted as needed until the same temperature consistently produced an NPS rating of approximately 60 in three separate trials. This full calibration procedure typically requires around 15 minutes to be completed. Only after achieving such consistency was the temperature defined as the participant's pain-60, and it was used for the THP stimulation applied alone ("test-stimulus alone") and during CPM ("test-stimulus under conditioning stimulus", see below for further details). It should be noted that although a fixed stimulus intensity is used based on the determination of pain-60, which reflects the perceived pain intensity at the onset of stimulation, reported pain ratings typically vary over the course of the test sessions as a function of several factors, including stimulus duration and stimulation sequence [30,62]. Notably, higher pain-60 temperatures are associated with greater sensitization, whereas lower pain-60 temperatures tend to be associated with adaptation; however, this does not compromise the validity of the CPM measure, as CPM is calculated using a subtraction-based index that inherently accounts for adaptation effects.

**2.3.2.b. Pain ratings of the test-stimulus alone.** For the pain ratings of the test-stimulus alone, the thermode was attached to the volar surface of the forearm of the dominant hand. The temperature was increased at a rate of 2°C/sec, from 32°C to the participant's pain-60 temperature, and remained fixed for 60 seconds. Participants were requested to rate perceived pain intensity on the NPS when the thermode reached the target temperature, and every 10 seconds

thereafter. The test-stimulus alone pain rating value was calculated by averaging the seven ratings obtained during the one-minute heat stimulation.

**2.3.2.c. Conditioned pain modulation (CPM).** CPM was assessed using a standardized protocol [11,63–65]. Hot water conditioning was selected over cold water to avoid sympathetic activation [66], potentially confounding the assessment of pain-related autonomic responses. Following recommendations from the literature [65,67], and evidence indicating the CPM effect is observed only when hot water immersion leads to moderate or intense pain [67], we adopted an individualized calibration procedure to ensure that the conditioning stimulus reliably elicited the necessary moderate to intense pain level across all participants (i.e., 30–70 in the NPS).

For the hot-water pain calibration, participants first immersed their non-dominant hand in a hot water bath set to 45.5 °C (Heto Cooling Bath CBN 8–30, Denmark). After 20 seconds of immersion, they were asked to rate their pain on a 0–100 NPS. If the rating fell within the target range of 30–70, no further adjustments were made. If the rating exceeded or fell below this range, the water temperature was adjusted by 0.5–1 °C in the appropriate direction, and the procedure was repeated. This iterative calibration continued until the participant reported a pain rating within the desired range. This individualized approach was implemented to minimize floor or ceiling effects in the conditioning stimulus and to ensure comparable subjective pain intensity across participants. In our sample, water temperature ranged between 44–47°C [mean temperature of 45.24°C (*0.74*)]. In the subsequent CPM procedure, participants immersed their non-dominant hand in the water bath with the calibrated heat temperature eliciting mild-to-moderate pain, and were asked to rate pain intensity again on the NPS after 20 seconds. Then, they were instructed to turn their attention to the dominant hand while keeping the non-dominant hand immersed. The test heat stimulus was applied for 60 seconds to the forearm of the dominant hand, during which participants were asked to rate the pain intensity every 10 seconds. After the heat pain termination, participants rated the hot water pain again before removing their hand. *CPM magnitude* was calculated by subtracting the mean pain ratings of the test-stimulus given alone from the mean test ratings under the conditioning stimulus, such that a lower (more negative) CPM magnitude score represented better pain inhibition, as recommended [65].

**2.3.3. Psychological traits and emotional state evaluation.** Catastrophic thinking about pain was assessed using *the Pain Catastrophizing Scale (PCS)* [68]*,* which includes 13 items rated on a 5-point scale across rumination, magnification, and helplessness. We used a validated Hebrew version [69]. Cronbach's alpha value in the current sample was 0.93. Fear of pain was measured with the *Fear of Pain Questionnaire (FPQ-9)* [70], consisting of nine items assessing fears of severe, minor, and medical/dental pain, rated on a 5-point scale. We translated and validated the questionnaire using a forward-backward method, based on [71]. Cronbach's alpha in the current sample was 0.85.

General state distress was assessed using *The Short Depression, Anxiety, and Stress Scale (DASS-21),* a set of three self-report scales measuring negative emotional states – depression, anxiety, and stress [72]. The scale contains 21 components rated on a 4-point scale according to the extent to which participants have experienced each state over the past week, with a total score calculated by summing up all items. We used a validated Hebrew version [73], and Cronbach's alpha value in the current sample was 0.91.

**2.3.4. Electrocardiography (ECG).** The signal was acquired using a BIOPAC MP150 data acquisition module (BIOPAC Systems, Inc., Goleta, CA, USA), at a sampling rate of 1000 HZ. Electrodes were applied in a three-lead, chest-mounted configuration with one electrode under the right clavicle and the other two electrodes on the lower right and left rib cages (i.e., lead II). During the recordings, participants were asked to sit quietly and to breathe naturally. A *resting* ECG was recorded for 5 minutes at the beginning of the experiment. ECG was also recorded 60 seconds before (*baseline*), during (*reactivity*), and after (*recovery*) THP given alone and during the CPM paradigm.

## 2.4. Data pre-processing

Normality was assessed for all variables using visual inspection via histogram plots and the Shapiro-Wilk normality test. The DASS scores showed left-skewed distribution, a pattern expected in a healthy, non-clinical sample, as emotional

distress measures often display skewed distributions in the general population (see, for example [74,75]). Given the sample size (>50) and the robustness of linear regression to mild non-normality in predictor variables, no transformations were applied, and the original DASS values were retained. Descriptive statistics were obtained for all study variables, and multicollinearity checks confirmed negligible overlap among questionnaire scores (Tolerance>0.10, VIF<5 [76];).

**2.4.1. Attentional bias tasks.** In both tasks, only RTs of correct trials containing a target were analyzed. RTs 2.5 z-scores above or below each participant's average RT in a specific condition (i.e., each combination of load and valence in the Perceptual Load task, and pain-congruent and pain-incongruent trials in the Dot-probe task) were removed from the analysis. This resulted in the removal of less than 1% from all trials, with an average of 3.5 trials per subject for both tasks. Participants whose average reaction times deviated more than ±2.5 standard deviations from the group mean were excluded from analysis, in order to remove cases with atypical overall performance that could reflect inattention, misunderstanding of the task, or other non-representative behavior. Using this criterion, data from one participant were excluded from the Perceptual Load task. Importantly, RTs in both tasks were normally distributed, supporting the assumptions required for parametric analyses.

Accuracy rates were also examined to identify deviant responses. In the Perceptual Load task, no participant showed systematically low accuracy levels in both the low and the high conditions. In the Dot-probe task, two participants consistently showed low accuracy levels (i.e., less than 0.8), and their task performance was excluded.

**2.4.2. ECG signal.** ECG pre-processing was conducted using the Python-based package Neurokit2 0.2.7 [77] and included a low-pass filter at 5 Hz and a notch filter at 50 Hz. According to published guidelines [78,79], QRS complexes were detected based on the steepness of the absolute gradient of the ECG signal. Subsequently, R-peaks were detected as local maxima in the QRS complexes. Artifacts were detected and corrected based on beat classification while removing cubic trends from the R-R interval series. The resulting R-R intervals were used to calculate the mean heart rate (HR) for all measured time phases (i.e., 5-minute rest HR in the one-minute baseline, reactivity, and recovery of THP given alone and during CPM). Mean HR (beats/min; bpm) measures were normally distributed in our sample; thus, the absolute values were used for the analysis. To account for individual differences in baseline HR, HR reactivity was expressed as percentage change scores [HR% change reactivity = (reactivity HR – baseline HR)/ baseline HR].

## 2.5. Statistical analysis

Statistical analyses were performed with SPSS version 27 (IBM Corp., Armonk, NY, United States) and R Statistical Software (v4.1.2; R Core Team 2021), using the interaction [80], yarr [81], and lavaan [82] packages.

**2.5.1. Paradigms assessment.** First, to examine whether the perceptual load manipulation affected participants' performance, a t-test was performed comparing accuracy rates between low and high load conditions, expecting higher error rates under high Perceptual Load. Next, a 2×2 repeated-measures ANOVA was conducted on RTs with load (low/high) and picture valence (pain/neutral) as within-subject factors to assess interference from pain images, with planned contrasts to test whether pain interference (i.e., longer RTs) emerges in both load conditions. Based on previous studies, we hypothesized that at the group level, interference from pain-distracting images will emerge only under the low load condition, when enough attentional resources are available [56–58]. Finally, to assess selective attention to pain words in the Dot-probe task, RTs were compared between pain-congruent and pain-incongruent trials. Faster RTs in pain-congruent trials were expected, indicating an attentional bias toward pain-related stimuli.

The CPM paradigm was evaluated by examining changes in pain ratings (i.e., at stimulus onset and after 10, 20, 30, 40, 50, and 60 seconds) during test-stimulus alone and under conditioning, using a two-way repeated measures ANOVA. Pain ratings were expected to be significantly lower under conditioning than when the test-stimulus is applied alone.

Before examining the associations between attentional biases, HR, and CPM, we first aimed to understand HR changes in relation to CPM. We therefore conducted a 2×3 repeated-measures ANOVA with paradigm (test-stimulus alone vs. test-stimulus under conditioning) and HR measurement stage (at baseline, during one-minute pain reactivity

and during one-minute pain recovery) as within-subject factors. Pain-60 temperature and mean conditioning pain ratings (average of water pain ratings at the start and end of the CPM paradigm) were included as covariates to control for their influence [30,62]. We expected HR to increase during pain (reactivity phase) and decrease during recovery from pain.

**2.5.2. Hypothesis testing.** Prior to the main regression analyses examining the relationships between our predictors and outcome measures, we conducted bivariate correlations (with Bonferroni correction) between the psychological traits, emotional-state measures, and attentional bias scores. This preliminary analysis enabled us to explore the associations among these variables and assess potential multicollinearity before including them in the regression models. Model fit for all regression analyses was evaluated using the Durbin–Watson statistic to assess autocorrelation of residuals, as recommended for linear models.

To evaluate whether attentional biases, psychological factors, and autonomic reactivity independently predicted pain sensitivity and pain modulation, two hierarchical multiple linear regressions were conducted, with pain-60 temperature and CPM magnitude serving as dependent variables. Predictors were entered in blocks to assess the incremental contribution of each variable set. In Block 1, attentional bias indices were entered (i.e., pain interference under low and high loads, and Dot-Probe attentional bias). Block 2 included the psychological traits and emotional-state measures (i.e., FPQ-9, PCS, and DASS), and Block 3 included the autonomic measure during pain (i.e., HR% reactivity change). To enhance robustness, bias-corrected and accelerated confidence intervals were calculated using 5,000 bootstrap repetitions [83].

## 3. Results

### 3.1. Participants' characteristics

Demographic information (age, BMI, dominant hand, and ethnicity), psychological traits and emotional state (Pain Catastrophizing Scale [PCS], Fear of Pain Questionnaire–Short Form [FPQ-9], and Depression Anxiety Stress Scales [DASS-21]), and baseline HR measures taken at 5 minutes rest are presented in Table 1. Due to a technical issue, the rest ECG recording of one participant was not obtained.

In this sample, BMI was not associated with HR at rest or during pain, and ethnicity did not significantly affect task performance (all $ps > .15$). These variables were therefore not included as covariates in subsequent analyses.

### 3.2. Paradigms assessment

**3.2.1. Interference from pain-related images in the Perceptual Load task.** There was a significant difference in accuracy between the load conditions of the Perceptual Load task, with higher accuracy in the low load condition (Mean accuracy: 0.94) than in the high load condition (Mean accuracy: 0.83; $t_{(85)}$=12.4, $p < .001$, Hedges' g = 1.34; see S2 Table). Similarly, a two-way repeated-measures ANOVA revealed longer RTs in the high load condition compared to the low load [$F_{(1,85)}$=3505.75, $p < .001$, $\eta^2 = .976$], confirming that the load manipulation effectively modulated task difficulty. A main effect of valence also emerged, showing slower RTs when pain-related interference images were presented versus neutral images [$F_{(1,85)}$= 6.186, $p = .015$ $\eta^2 = .068$]. The interaction between load and valence was not significant ($p = .74$). However, planned contrasts showed that, as expected, interference from pain-related images emerged under low load ($t_{(85)}$=3.074, $p = .003$, Hedges' g = 0.316, see Fig 4), but not under the high perceptual load ($p = .219$).

**3.2.2. Attention bias in the Dot-probe task.** The mean accuracy in identifying the probe letter in the task was 0.96, with no significant difference between pain-congruent and pain-incongruent trials ($p = .94$). Further, there was no difference in mean RTs to probes appearing in the pain-congruent and pain-incongruent locations (mean RTs: pain-congruent: 493.8 ms; pain-incongruent: 494.5 ms; $p = .75$, see S2 Table), indicating no group-level selective attention to pain.

**3.2.3. Pain ratings during test-stimulus alone and test-stimulus under conditioning.** The pain-60 temperatures used for the test-stimulus ranged between 41–48°C with a mean temperature of 45.04°C ($SD = 1.95$). The hot-water conditioning stimulus produced an average NPS rating of 53.89 ($SD = 16.02$). Pain ratings collected after the first 20

**Table 1. Participant Demographics and Psychological Characteristics.**

| | Total (N=86) |
|---|---|
| **Age (years)** | |
| Mean (SD) | 23.21 (3.0) |
| Range (Min-Max) | 18-33 |
| **Body Mass Index (BMI)** | |
| Mean (SD) | 22.31 (3.2) |
| Range (Min-Max) | 17- 32 |
| **Dominant Hand** | |
| Right hand (%) | 79 (91.9%) |
| Left Hand (%) | 7 (8.1%) |
| **Ethnicity** | |
| Jews (%) | 49 (57%) |
| Arabs (%) | 37 (43%) |
| **Fear of Pain (FPQ-9; range 9–45)** | |
| Mean (SD) | 25.23 (6.5) |
| Range in the current sample (Min-Max) | 9- 40 |
| **Pain catastrophizing (PCS; range: 0–52)** | |
| Mean (SD) | 17.35 (10.5) |
| Range in the current sample (Min-Max) | 0- 42 |
| **General distress (DASS-21; range: 0–63)** | |
| Mean (SD) | 11.73 (9.4) |
| Range in the current sample (Min-Max) | 0- 46 |
| **HR[a] at baseline 5 minutes rest (bpm)** | |
| Mean (SD) | 82.0 (11.3) |
| Range in the current sample (Min-Max) | 62-113 |

[a]HR = Heart rate.

seconds of hand immersion averaged 44.55 NPS (SD = 16.57) and significantly increased over the 2.5-minute immersion period, with the final rating averaging 63.23 NPS (SD = 20.80; $t_{(83)}$ = 8.56, p < .001, Hedges' g = 0.936).

In both paradigms, pain ratings significantly decreased over the one-minute stimulation period ($F_{(2,179)}$=91.85, p < .001, $\eta^2$ = .52). Overall, the results indicate a main effect of pain paradigm, as pain ratings were lower for the test-stimulus under conditioning compared to the test-stimulus given alone ($F_{(1,85)}$=5.89, p = .017, $\eta^2$ = .065). There was also a significant paradigm × time interaction ($F_{(5,363)}$=3.4, p = .008, $\eta^2$ = .039). Bonferroni-corrected post-hoc analysis revealed no differences between paradigms in the first 20 seconds (all ps > .2), but from 30 seconds onward, pain ratings were significantly lower for test-stimulus under conditioning (p < .05; See Table 2). These findings confirm the effective engagement of the descending pain inhibition and reliable time-dependent pain reduction.

### 3.3. HR changes during the test-stimulus alone and under conditioning

HR and HR %change at the one-minute baseline, reactivity, and recovery of the test-stimulus alone and under conditioning are presented in Table 3.

The repeated-measures ANOVA assessing HR before, during, and at recovery from the test-stimulus alone and under conditioning revealed a significant main effect of paradigm, with higher HR during test-stimulus under conditioning compared to the test-stimulus alone [$F_{(1,74)}$=27.54, p < .001, $\eta^2$ = .271]. There was no main effect of experimental

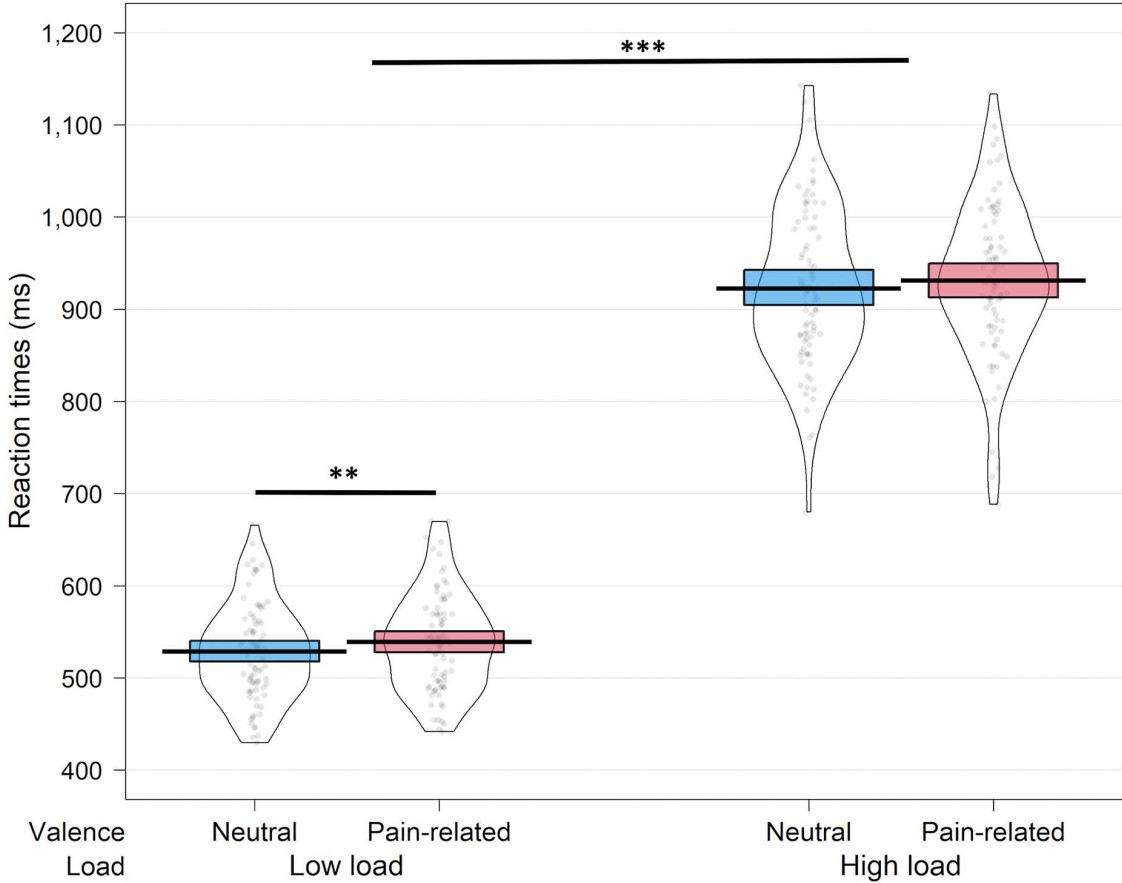

**Fig 4. Mean reaction times (RTs) in the Perceptual Load task as a function of load and picture valence.** RTs were generally slower under high perceptual load. Under low load, pain-related interfering images significantly slowed target identification compared to neutral images, whereas no such effect was observed under high load. *Note: The RT scale is set from 400 to 1,200 ms to facilitate visualization of condition differences. Dots represent individual participants, the black line indicates the mean, and the shaded area represents the 95% confidence interval (CI). \*\*p ≤ .01, \*\*\*p ≤ .001.*

stage [$F_{(2,73)}$=0.97, $p$=.257]; however, a significant paradigm × experimental stage interaction emerged [$F_{(1,73)}$=7.22, $p$=.001, $\eta^2$=.17; see Fig 5]. Bonferroni-corrected post-hoc analyses indicated that HR increased significantly from baseline to reactivity during pain stimulation only when the test-stimulus was under conditioning [mean difference=5.35 bpm; 95% CI, 4.2 to 6.5; $p$=.009, Hedges' g=1.24]. Moreover, the HR during test-stimulus under conditioning was significantly higher than the HR during the test-stimulus alone [mean difference=5.2 bpm; 95% CI, 4.4 to 6.0, $p$=.009, Hedges' g=1.49].

During recovery from the test-stimulus alone, HR remained elevated compared to baseline [mean difference=1.48 bpm; 95% CI, 0.605 to 2.3; $p$=.009, Hedges' g=0.46]. The same pattern of heightened HR compared to baseline was observed during the recovery from the test-stimulus under conditioning [mean difference=4.47 bpm; 95% CI, 3.32 to 5.6, $p$=.009, Hedges' g=1.02]. Notably, during recovery, HR was significantly higher after the test-stimulus under conditioning compared to the test-stimulus alone [mean difference=3.24 bpm; 95% CI, 2.43 to 4.1, $p$=.009, Hedges' g=0.87].

The influence of pain-60 temperature on HR reactivity was insignificant [$F_{(7,74)}$=1.96, $p$=.072], while higher pain ratings of the hot water resulted in higher HR elevations during the test-stimulus under conditioning [$F_{(1,74)}$=13.26, $p$<.001, η2=.152].

**Table 2. Mean and SD of pain intensity ratings on the 0-100 NPS along the one-minute test heat pain alone and under conditioning.**

| Pain rating condition | At stimulus onset | After 10 seconds | After 20 seconds | After 30 seconds | After 40 seconds | After 50 seconds | After 60 seconds | Mean rating |
|---|---|---|---|---|---|---|---|---|
| **Test-stimulus alone** | 59.41 (*18.8*) | 36.26 (*21*) | 25.98 (*21.5*) | 25.93 (*23.4*) | 26.31 (*25.7*) | 25.13 (*26.2*) | 25.63 (*26.3*) | 30.86 (*16.6*) |
| **Test-stimulus under conditioning** | 61.62 (*18.7*) | 33.05 (*21.9*) | 25.40 (*21.85*) | 21.44 (*22*) | 21.41 (*22.5*) | 20.06 (*23.4*) | 19.93 (*23.7*) | 28.98 (*17.9*) |
| | $t_{(85)}$= 1.16 $p$=.248 Hedges' g=0.135 | $t_{(85)}$=1.29 $p$=.198 Hedges' g=0.099 | $t_{(85)}$=0.28 $p$=.779 Hedges' g=0.003 | $t_{(85)}$=2.52 $p$=.014 Hedges' g=0.268 | $t_{(85)}$=2.80 $p$=.006 Hedges' g=0.295 | $t_{(85)}$=2.91 $p$=.004 Hedges' g=0.304 | $t_{(85)}$=3.30 $p$=.001 Hedges' g=0.327 | $t_{(85)}$=2.06 $p$=.043 Hedges' g=0.191 |

Note: T-test values represent the difference between test-stimulus pain rated alone and pain under conditioning, with Bonferroni correction for multiple comparisons.

**Table 3. Descriptive statistics of HR and %HR during the test-stimulus given alone and during conditioning.**

| Paradigm | HR[a] at one-minute baseline | HR[a] during one-minute pain reactivity | %HR change reactivity[b] | HR[a] during one-minute pain recovery | %HR change recovery[c] |
|---|---|---|---|---|---|
| **Test-stimulus pain alone** Mean (SD) Range | 77.24 (*10.6*) [54.9 - 99.9] | 77.72 (*10.9*) [57.8 - 103.7] | 0.7% (*5.1*) [-7.9 - 14.8] | 78.78 (*10.3*) [57.9-104.2] | 1.6% (*5.1*) [-11.2 - 21.1] |
| **Test-stimulus under conditioning** Mean (SD) Range | 77.54 (*10.7*) [57.8 - 101.4] | 82.90 (*10.6*) [59.4 - 110.9] | 7.2% (*6.2*) [-2.9 - 19.3] | 82.02 (*10.2*) [61.5-111.6] | −1.0% (*5.2*) [-13.6 - 11.3] |

[a]HR = Heart rate;

[b]%HR change reactivity = (HR during reactivity − baseline HR)/ baseline HR; positive values indicate an HR increase in response to pain;

[c]%HR change recovery = (HR during recovery − HR during reactivity)/ HR during reactivity; positive values indicate higher HR during recovery than reactivity;

### 3.4. Correlations between the examined predictors

Bivariate correlation analysis was conducted between the psychological traits, emotional state, attentional bias indexes, baseline HR, and HR reactivity to explore the associations among these variables and assess potential multicollinearity before including them in the regression models. This analysis revealed that interference from pain-related images under both perceptual load conditions was not significantly associated with any psychological traits or emotional states (see S3 Table in the supplementary materials). In contrast, the dot-probe attention bias was negatively correlated with fear of pain, as assessed by the FPQ-9 (r=−0.27, $p$=.039), indicating that greater attentional avoidance of pain-related words was associated with higher fear of pain. The psychological trait and emotional state questionnaires showed moderate intercorrelations, and no indication of multicollinearity was observed.

HR at baseline did not correlate with any of the other predictors. However, greater attention bias in the dot-probe was linked to a stronger autonomic response during the CPM task (r=0.29, $p$=.032). Moreover, state distress levels were negatively correlated with HR response during the test-stimulus under conditioning (r=−0.29, $p$=.004), indicating that higher distress was associated with diminished HR responsiveness.

### 3.5. Predicting pain sensitivity and pain modulation

#### 3.5.a. Pain sensitivity evaluated by pain-60 temperature.
The results of the block hierarchical multiple regression predicting pain sensitivity, as measured by pain-60 temperature, are presented in Table 4. Block 1, which included

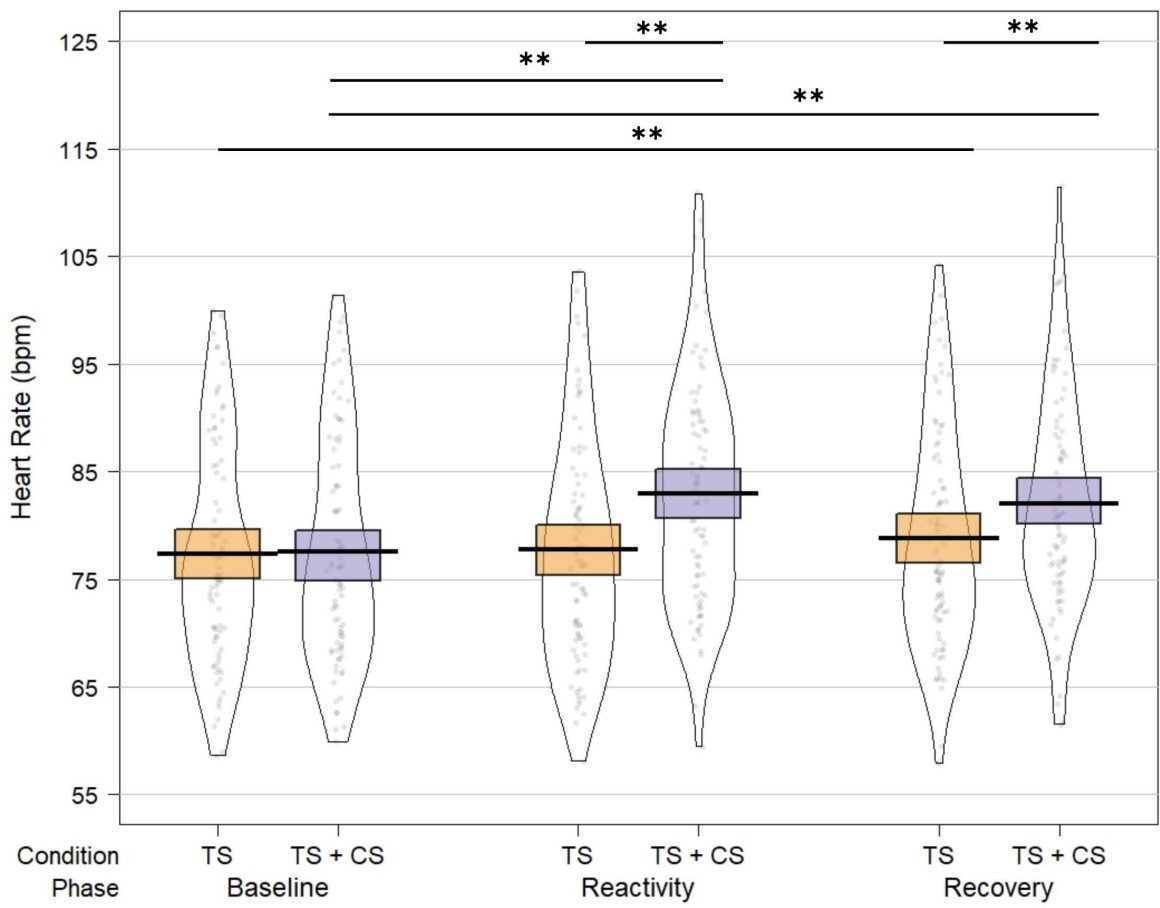

**Fig 5. HR at baseline, reactivity, and recovery during the test-stimulus alone (TS) and under conditioning (TS+CS).** HR was higher during the reactivity and recovery phases of the test-stimulus under conditioning compared to the corresponding phases of the test-stimulus alone. Moreover, HR during the recovery phase remained elevated relative to baseline in both conditions. *Note: For clarity in visualizing differences between conditions, the HR scale is set from 55 to 125 bpm. Dots represent individual participants, the black line indicates the mean, and the shaded area represents the 95% confidence interval (CI). **p ≤ .01; HR = heart rate.*

attentional bias measures, explained 10.4% of the variance in pain-60 temperature, with a standard error of estimate of 1.89 [$F_{(3,76)}$=2.94, $p$=.035, $R^2$=.104, Adjusted $R^2$=0.07, SEE=1.89]. Specifically, higher pain interference under both low (β=−0.19, 95% BCa CI [−0.025, −0.002], $p$=.020) and high perceptual loads (β=−0.25, 95% BCa CI [−0.017, −0.002], $p$=.015) predicted lower temperatures rated as pain-60, indicating greater pain sensitivity. Collinearity diagnostics indicated no multicollinearity concerns (VIFs<1.2), the residuals were approximately normally distributed and homoscedastic, with no autocorrelation (Durbin–Watson=2.20). Standardized residuals were within ±3 SD, indicating no extreme outliers.

Adding psychological variables in Block 2 and baseline HR in Block 3 did not significantly improve the model. The full model, including all predictors, was not significant [$F_{(7,72)}$=1.42, $p$=.213, $R^2$=.121, Adjusted $R^2$=0.035], with the additional blocks not producing a significant increase in explained variance [$\Delta R^2$=.017, $F_{(4,72)}$=4.29, $p$=.733]. These results indicate that the relationship between attentional bias and pain sensitivity was not accounted for by individual differences in psychological variables, suggesting that attentional bias independently contributes to the prediction of pain sensitivity.

Table 4. Block multiple linear regression analysis predicting pain-60 temperature.

| Model | Predictor | B | β | 95% BCa CI | p predictor[a] | R² | Adjusted R² | ΔR² change[b] | SEE[c] | F change[d] (df) | p model change[d] |
|---|---|---|---|---|---|---|---|---|---|---|---|
| 1 | **Attention bias Dot-Probe** | −0.006 | −0.070 | [−0.028, 0.014] | .503 | 0.104 | 0.07 | – | 1.89 | – | – |
| | **Pain interference – Low Perceptual Load** | 0.012 | −0.195 | [−0.025, -0.002] | ***.020*** | | | | | | |
| | **Pain interference – High Perceptual Load** | −0.008 | −0.251 | [−0.011, -0.002] | ***.015*** | | | | | | |
| 2 | **FPQ-9[e]** | −0.036 | −0.121 | [−0.126, 0.047] | .378 | 0.119 | 0.05 | .018 | 1.91 | 0.43 $_{(3, 73)}$ | .734 |
| | **PCS[f]** | −0.006 | −0.031 | [−0.064,0.035] | .778 | | | | | | |
| | **DASS[g]** | 0.022 | 0.104 | [−0.034, 0.078] | .620 | | | | | | |
| 3 | **Baseline HR[h]** | 0.007 | 0.038 | [−0.027, 0.039] | .684 | 0.121 | 0.035 | 0.001 | 1.92 | 0.12 $_{(1, 72)}$ | .732 |

[a]Significant p-values are presented in bold, based on bias-corrected and accelerated (BCa) 95% confidence intervals derived from 5,000 bootstrap samples.

[b]ΔR² indicates the change in R² from the previous model step, reflecting the additional variance explained by the newly added predictors.

[c]SEE = Std. Error of the Estimate.

[d]F change and p (model change) refer to the statistical significance of the increase in explanatory power after adding these predictors.

[e]FPQ-9 = Fear of Pain Questionnaire – 9-item version.

[f]PCS = Pain Catastrophizing Scale.

[g]DASS = Depression, Anxiety, and Stress Scale – Short Form.

[h]baseline HR = Heart rate during 5-minutes rest at the beginning of the experiment.

**3.4.b. Pain modulation evaluated by CPM magnitude.** The block hierarchical multiple regression predicting pain modulation, while controlling for test-stimulus intensity (i.e., pain-60 temperature, entered as a covariate of no interest in the first block), did not predict CPM magnitude [$F_{(8,71)}$=1.58, $p$ = .145, R² = .151, Adjusted R² = 0.056, SEE = 11.2]. None of the individual predictors contributed significantly to the model (see Table 5), with no evidence of multicollinearity (VIFs < 1.4). Residuals were approximately normally distributed and homoscedastic, with no evidence of autocorrelation (Durbin–Watson = 1.97).

## 3.6. Exploratory analysis

Given that our results showed elevated HR levels during pain reactivity to test-stimulus under conditioning, which remained high during the recovery period, we were interested in investigating whether attentional bias, psychological traits, and emotional state predict the level of HR changes in both of these stages.

To account for individual differences in both HR reactivity and recovery, we used the previously computed HR% change reactivity (see the Method section), and computed a new percentage-based recovery index [HR% change recovery = (recovery HR – reactivity HR)/reactivity HR].

In our exploratory analysis, as was done in our main multiple linear regression analyses, the predictors were entered hierarchically in blocks to examine their incremental contribution [Block 1: Pain-60 temperature (i.e., to control for the thermal stimulus intensity that can affect autonomic reactivity), Block 2: Attentional bias measures (i.e., attention bias in the low and high load conditions of the Perceptual load task, and attention bias in the Dot-Probe task), Block 3: Psychological variables (i.e., FPQ-9, PCS, DASS)]. To increase the robustness of the results, bias-corrected and accelerated confidence intervals were calculated using 5,000 bootstrap samples.

**3.5.a. Predicting %HR change during reactivity to the test-stimulus under conditioning.** The multiple regression model predicting %HR change during test-stimulus under conditioning was significant and explained 21.4% of the variance in

**Table 5. Multiple linear regression analysis predicting CPM magnitude.**

| Model | Predictor | B | β | 95% BCa CI | p predictor[a] | R² | Adjusted R² | ΔR² change[b] | SEE[c] | F change[d] (df) | p model change[d] |
|---|---|---|---|---|---|---|---|---|---|---|---|
| 1 | Pain-60 temperature | −0.183 | −0.031 | [-1.70, 1.33] | .792 | 0.01 | 0.013 | − | 11.61 | − | − |
| 2 | Attention bias to pain-related words | −0.096 | −0.192 | [-0.230, 0.029] | .121 | .086 | 0.04 | .086 | 11.31 | 2.35 (3,75) | 0.079 |
| | Interference from pain-related images in the low perceptual load condition | −0.007 | −0.017 | [-0.102, 0.070] | .884 | | | | | | |
| | Interference from pain-related images in the high perceptual load condition | −0.047 | −0.217 | [-0.096, 0.001] | .059 | | | | | | |
| 3 | FPQ-9[e] | −0.126 | −0.071 | [-0.627, 0.372] | .577 | 0.113 | .0.027 | .027 | 11.37 | 0.74 (3,72) | .432 |
| | PCS[f] | 0.109 | −0.099 | [-0.201, 0.443] | .451 | | | | | | |
| | DASS[g] | −0.307 | −0.243 | [-0.649, 0.009] | .065 | | | | | | |
| 4 | %HR change CPM reactivity[h] | −43.6 | −0.222 | [-94.28, 7.93] | .078 | 0.151 | 0.056 | .038 | 11.2 | 3.20 (1,71) | .078 |

[a]Significant p-values are presented in bold, based on bias-corrected and accelerated (BCa) 95% confidence intervals derived from 5,000 bootstrap samples.

[b]ΔR² indicates the change in R² from the previous model step, reflecting the additional variance explained by the newly added predictors.

[c]SEE = Std. Error of the Estimate.

[d]F change and p (model change) refer to the statistical significance of the increase in explanatory power after adding these predictors.

[e]FPQ-9 = Fear of Pain Questionnaire – 9-item version.

[f]PCS = Pain Catastrophizing Scale.

[g]DASS = Depression, Anxiety, and Stress Scale – Short Form.

[h]*%HR change during CPM reactivity = the difference between HR during the test-stimulus under conditioning (CPM) and HR at baseline before the test, divided by the baseline HR.*

%HR change, with a standard error of estimate of 0.054 [$F_{(7,73)}$=2.84, *p*=.011, R²=0.214, Adjusted R²=.139, SEE=0.054, Table 6]. Among the predictors, only general distress (DASS-21) was significant. Higher distress levels were associated with a smaller HR change during CPM reactivity (β=−0.310, 95% BCa CI [−0.004, −0.001], *p*=.012; see Fig 6A). Collinearity diagnostics indicated no multicollinearity concerns (VIFs<1.4), the residuals were approximately normally distributed and homoscedastic, with no autocorrelation (Durbin–Watson=1.5). Standardized residuals were within ±2 SD, indicating no extreme outliers.

**3.5.b.  HR %change during recovery from CPM.**  The multiple regression model explained 31.2% of the variance in %HR changes during recovery for pain test-stimulus under conditioning [$F_{(7,73)}$=4.73, *p*<.001, R²=.312, Adjusted R²=.246, SEE=0.045]. As shown in Table 7, only attentional bias variables significantly predicted HR changes during recovery from pain. Specifically, higher attentional avoidance in the Dot-Probe task (β=−0.334, 95% BCa CI [−0.001, −0.000], *p*=.003; see Fig 6B) and greater interference under low perceptual load (β=0.224, 95% BCa CI [0.000, 0.001], *p*=.017; see Fig 6C) were associated with increased HR during pain recovery relative to reactivity. No other predictors were significant. No autocorrelation emerged (Durbin–Watson=2.07), with standardized residuals indicating no extreme outliers.

## 4.  Discussion

This study investigated whether attentional bias predicts individual differences in pain sensitivity and endogenous pain modulation, and whether psychological, emotional, and autonomic factors provide additional explanatory value. The findings suggest that greater interference from pain-related images in a novel Perceptual Load task was associated with

**Table 6. Multiple linear regression analysis predicting HR %change during CPM.**

| Model | Predictor | B | β | 95% BCa CI | p predictor[a] | R² | Adjusted R² | ΔR² change[b] | SEE[c] | F change[d] (df) | p model change[d] |
|---|---|---|---|---|---|---|---|---|---|---|---|
| 1 | **Pain-60 temperature** | 0.004 | 0.129 | [-0.002, -0.010] | *.184* | 0.018 | .005 | — | 0.058 | — | — |
| 2 | **Attention bias Dot-Probe** | 0.001 | 0.212 | [-0.000, -0.001] | *.085* | 0.110 | .063 | .09 | 0.056 | 2.61 (3, 76) | .057 |
| | **Pain interference -Low Perceptual Load** | 0.000 | −0.083 | [0.000, 0.001] | *.417* | | | | | | |
| | **Pain interference – High Perceptual Load** | −0.001 | −0.017 | [-0.001, 0.000] | *.874* | | | | | | |
| 3 | **FPQ-9[e]** | −0.001 | −0.007 | [-0.002, 0.002] | *.954* | .214 | .139 | .11 | 0.054 | 3.24 (3, 73) | ***.027*** |
| | **PCS[f]** | 0.000 | −0.041 | [-0.002, 0.001] | *.742* | | | | | | |
| | **DASS[g]** | −0.002 | −0.310 | [-0.004, -0.001] | ***.012*** | | | | | | |

[a]Significant p-values are presented in bold, based on bias-corrected and accelerated (BCa) 95% confidence intervals derived from 5,000 bootstrap samples.

[b]ΔR² indicates the change in R² from the previous model step, reflecting the additional variance explained by the newly added predictors.

[c]SEE = Std. Error of the Estimate.

[d]F change and p (model change) refer to the statistical significance of the increase in explanatory power after adding these predictors.

[e]FPQ-9 = Fear of Pain Questionnaire – 9-item version.

[f]PCS = Pain Catastrophizing Scale

[g]DASS = Depression, Anxiety, and Stress Scale – Short Form.

heightened pain sensitivity. However, no significant associations were found between attentional biases or other examined predictors and the efficacy of pain inhibition.

An exploratory analysis further investigated whether attentional biases, psychological traits, and emotional state predict HR reactivity and recovery during CPM. Higher psychological distress (DASS-21) was associated with reduced HR reactivity, indicating a blunted autonomic response under the CPM procedure. In contrast, greater attentional biases—reflected by pain-cue avoidance in the Dot-Probe task and increased pain interference under low load—predicted higher HR during recovery, suggesting delayed autonomic recovery linked to pain-related attentional biases. Taken together, these findings demonstrate that higher attention biases towards pain-related information predict stronger pain sensitivity and delayed recovery from painful stimulation, highlighting the relation between individual differences in orienting attention towards pain and pain reactivity.

## 4.1. Attentional biases as predictors of pain sensitivity

Empirical research has yielded mixed findings regarding attentional biases toward pain, highlighting the complexity of these attentional patterns across populations and contexts [10]. Recent meta-analyses report increased attention to painful stimuli in chronic pain patients [23,84,85], while others suggest that attentional avoidance from pain cues is linked to pain chronicity and postoperative pain [86–88]. Notably, in healthy individuals, attentional biases to pain are often not detected using traditional RT-based tasks, but may emerge in more sensitive paradigms, such as eye-tracking [23,85,89]. In the current study, we employed two tasks: a classical Dot-Probe task and a novel Perceptual Load task, to capture different aspects of attention tendencies towards pain-related stimuli. Consistent with previous findings [23], the Dot-Probe task revealed no group-level attentional bias towards pain cues. In contrast, the Perceptual Load task revealed attentional interference from pain-related images under low perceptual load, indicating that RT-based measures can detect subtle attentional biases in healthy individuals when cognitive resources allow processing of both distracting and task-relevant stimuli. This suggests that the Perceptual Load task may offer greater sensitivity than traditional RT paradigms, without requiring eye-tracking methods.

## CPM HR Reactivity

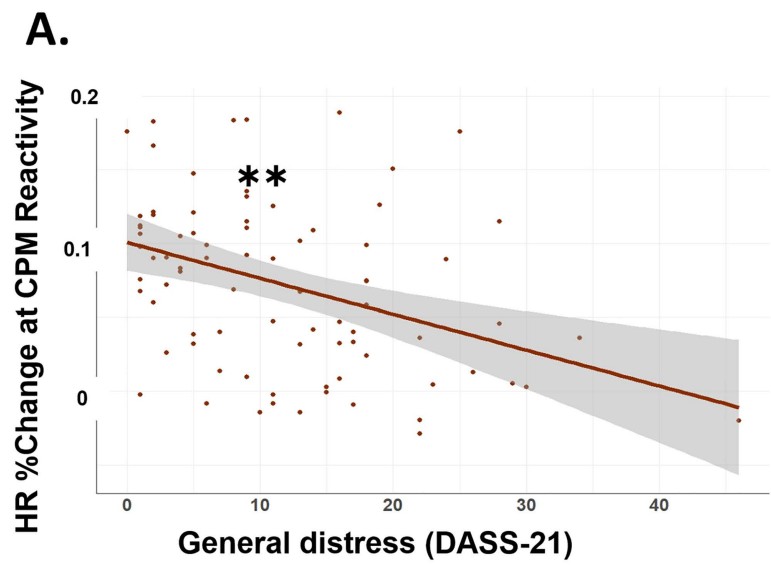

## CPM HR Recovery

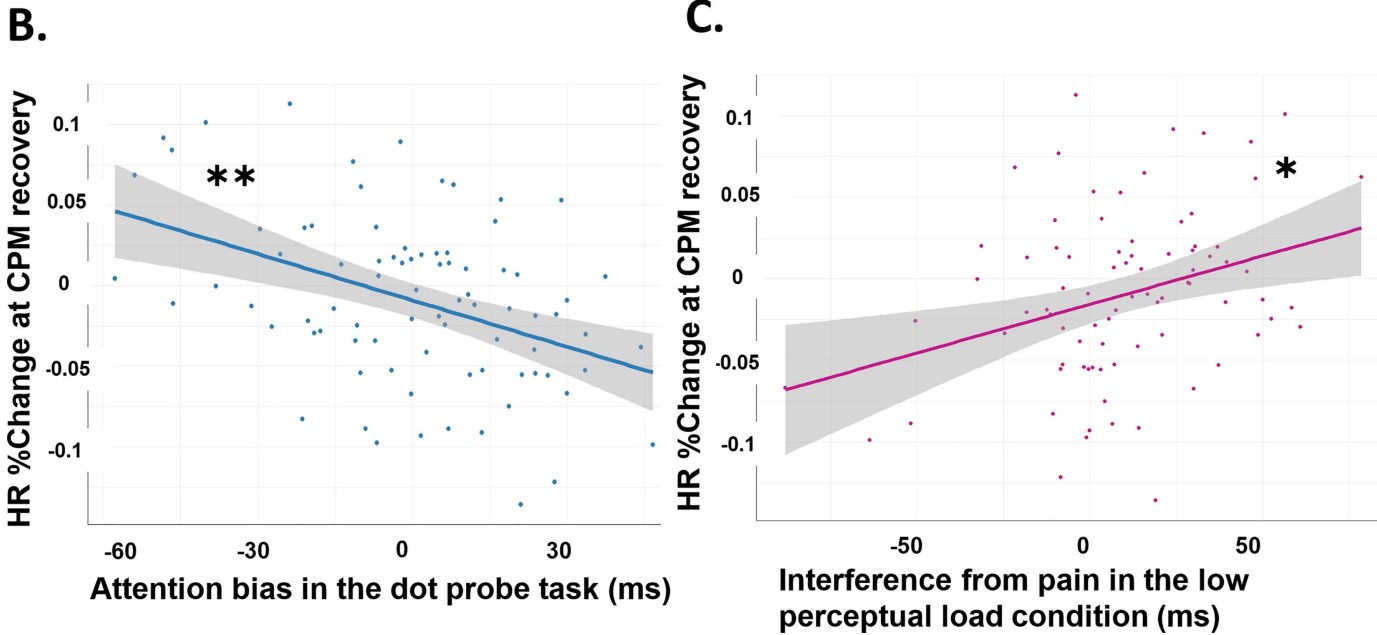

**Fig 6. Results of the exploratory analysis predicting HR reactivity (panel A) and recovery (panels B, C) from the CPM paradigm. A.** Higher levels of general distress predicted less change in HR during one-minute reactivity to the test-stimulus under conditioning. **B.** Higher attentional avoidance from pain-related words predicted elevated HR during recovery from the test-stimulus under conditioning. **C.** Interference from pain-related images under the low cognitive load of the Perceptual Load task predicted elevations in HR during recovery from the test-stimulus under conditioning. *Note. CPM %HR change reactivity = the difference between HR during the test-stimulus under conditioning (CPM reactivity) and HR at baseline before the test, divided by the baseline HR; CPM %HR change at recovery = the difference between HR during the test-stimulus under conditioning (CPM reactivity) and during recovery, divided by the HR at reactivity. Dots represent individual participants, and the shaded area represents the 95% confidence interval (CI) *p < .05; **p < .01.*

**Table 7. Multiple linear regression analysis predicting HR %change during recovery from CPM paradigm.**

| Model | Predictor | B | β | 95% BCa CI | p pre-dictor[a] | R² | Adjusted R² | ΔR² change[b] | SEE[c] | F change[d] (df) | p model change[d] |
|---|---|---|---|---|---|---|---|---|---|---|---|
| 1 | **Pain-60 temperature** | −0.002 | −0.059 | [-0.007, -0.004] | *.124* | 0.016 | .004 | — | 0.052 | — | — |
| 2 | **Attention bias Dot-Probe** | −0.001 | −0.334 | [-0.001, -0.000] | ***.003*** | 0.254 | .215 | .24 | 0.046 | 8.08 (3, 76) | ***<.001*** |
| | **Pain interference -Low Perceptual Load** | 0.001 | 0.224 | [0.000, 0.001] | ***.017*** | | | | | | |
| | **Pain interference – High Perceptual Load** | 0.000 | 0.093 | [-0.001, 0.002] | *.339* | | | | | | |
| 3 | **FPQ-9[e]** | 0.001 | 0.091 | [-0.001, 0.003] | *.416* | 0.312 | .246 | .06 | 0.045 | 2.06 (3, 73) | .113 |
| | **PCS[f]** | 0.000 | 0.087 | [-0.001,0.002] | *.456* | | | | | | |
| | **DASS[g]** | 0.001 | 0.151 | [-0.001, 0.003] | *.156* | | | | | | |

[a]Significant p-values are presented in bold, based on bias-corrected and accelerated (BCa) 95% confidence intervals derived from 5,000 bootstrap samples.

[b]ΔR² indicates the change in R² from the previous model step, reflecting the additional variance explained by the newly added predictors.

[c]SEE = Std. Error of the Estimate.

[d]F change and p (model change) refer to the statistical significance of the increase in explanatory power after adding these predictors.

[e]FPQ-9 = Fear of Pain Questionnaire – 9-item version.

[f]PCS = Pain Catastrophizing Scale.

[g]DASS = Depression, Anxiety, and Stress Scale – Short Form.

At the individual level, participants exhibited considerable variability in attentional bias indexes across the two tasks, which were not correlated with each other. Moreover, each task-specific attentional bias index was associated with a distinct research variable: avoidance of pain-related words in the Dot-Probe task was linked with greater fears of pain, whereas interference from pain-related images in the Perceptual Load task was associated with increased pain sensitivity.

This dissociation can be understood within contemporary models of attentional biases, which distinguish between automatic attentional processes (i.e., facilitated attention towards a threatening stimulus and difficulty disengaging from it) with more strategic ones (i.e., attentional avoidance) [90,91]. The short stimulus duration in the Perceptual Load task (i.e., 200ms) likely tapped early attentional capture, while the longer cue exposure in the Dot-Probe (i.e., 500ms) could assess later, more controlled attentional processes [92]. The distinct results from both tasks are consistent with previous studies showing that within a single experimental context, attentional bias patterns depend on the processing stage, with initial engagement and difficulties disengaging from pain cues being followed by later attentional avoidance [93,94]. Together, these findings underscore the value of employing multiple attention tasks or systematically manipulating cue presentation times to capture different components of pain-related attentional processing, each of which may differentially contribute to individual differences in pain perception and fear responses. This dynamic pattern also lends support to vigilance–avoidance models of attention in pain, which propose that attentional bias patterns shift as a function of perceived threat and individual fear levels [10,95].

At the same time, methodological differences between the tasks should also be considered when interpreting the observed patterns. Although both tasks used pain-related cues, the Perceptual Load task employed pain images, whereas the Dot-probe task used pain words. These differences in stimulus modality could also account for the divergent findings [96,97], as pain images are more ecologically valid and closely resemble real pain experiences, while pain words may instead activate verbal conceptualizations of pain, often captured by self-report questionnaires [10,98]. Indeed, previous meta-analyses have shown that stimulus type can influence the detection of attentional biases [23], with participants more likely to fixate on pain pictures than on neutral pictures, but not on pain-related words [85].

Stimulus characteristics and presentation times also affect the efficacy of Attention Bias Modification (ABM) paradigms using the Dot-Probe task, with training effects varying depending on the type of stimuli (e.g., words versus faces [28]; sensory or affective words [99]), and the duration of stimulus presentation [27]. Moreover, the two tasks differ in the relevance of distracting stimuli to the primary task. In the Perceptual Load task, pain-related images are unrelated to the target task and must be ignored, whereas in the Dot-Probe task, pain-related words cue participants' attention to the target letter and are therefore more closely integrated into task demands. These differences in task structure may further contribute to the inconsistencies observed across paradigms in our results [97].

## 4.2. Emotional predictors of pain sensitivity and pain modulation

Contrary to our hypotheses, none of the psychological traits or emotional states (i.e., fear of pain, pain catastrophizing, and emotional distress) directly predicted pain sensitivity. Fear of pain was, however, associated with attentional avoidance, yet it did not explain additional variance in pain sensitivity. This pattern suggests that, in healthy individuals, psychological traits may have only a limited direct influence on pain perception but could indirectly shape pain experiences through their effects on attentional processes [26,100]. These findings are consistent with a systematic review reporting weak and inconsistent associations between psychological traits and pain sensitivity [101]. It is also consistent with studies suggesting that psychological traits may influence pain primarily through interactions with situational factors, such as threat levels or current mood [102,103], and can modulate attentional biases toward pain-related stimuli [104–106].

Similarly, we did not find significant associations between these psychological variables and CPM. Although some works point out the contribution of psychological variables to CPM efficiency [32,33], others have shown that they explain only a small portion of the variance in CPM magnitude. For example, depression, anxiety, and pain catastrophizing accounted only for 3% of the variance in CPM magnitude [14] and were associated only with specific CPM pain modalities [31]. Taken together, these findings suggest that in healthy individuals, psychological traits and emotional states have limited influence on both pain sensitivity and endogenous pain modulation, which may become more pronounced in clinically relevant contexts.

## 4.3. Attentional biases towards pain do not affect pain inhibition efficiency

The role of attention in pain modulation, particularly in CPM, remains debated [13,16]. While attention has been proposed to influence CPM [65], findings from a recent meta-analysis are mixed [16]. CPM appears to be enhanced when attention is experimentally directed toward the conditioning stimulus and reduced when attention is directed toward the test stimulus. However, studies using external distraction tasks during CPM have produced inconsistent results [16].

Unlike the goal-directed attentional processes typically examined in CPM research, attentional biases reflect more automatic, involuntary tendencies to focus on pain-related cues, typically assessed in contexts where pain is incidental rather than goal-relevant [6,89]. Our findings add a novel perspective by showing that while attentional biases are linked with increased pain sensitivity, they do not predict CPM magnitude. One potential explanation is that CPM represents a dynamic, goal-directed modulation of pain processing that is less susceptible to automatic cognitive-affective traits in healthy individuals [14]. Neural evidence may support this distinction: attentional biases engage circuits involved in threat detection and affect regulation (e.g., amygdala, ACC, orbitofrontal cortex) [90,107,108], which likely influence early pain detection and sensitivity [100]. However, CPM is dependent on brainstem-driven descending modulation, which may be less impacted by attention traits and affect regulation in healthy individuals [109,110].

A further challenge in the literature is that "attention" is often treated as a unitary construct [1,2]. Without systematic comparisons across tasks, it remains unclear whether divergent findings reflect differences in attentional stages, as proposed by Todd et al. (2015), or instead arise from variation in task characteristics. One important distinction concerns whether attentional capture is assessed implicitly (e.g., reaction-time interference paradigms) or explicitly (e.g.,

self-reports of noticing or attending to specific stimuli) [111]. These approaches do not necessarily index the same attentional mechanisms and may be differentially influenced by cognitive control processes [111,112].

This distinction is particularly relevant when comparing our findings with recent work on intrinsic attention to pain, which reflects a more explicit, trait-like attentional measure [21]. Adams et al. (2021) assessed pain-related attentional tendencies by asking participants to report the extent to which their thoughts were focused on pain during brief noxious stimulation. They found that higher self-reported intrinsic attention to pain predicted both less efficient pain inhibition and greater pain amplification [21]. In contrast to this explicit approach, our tasks assessed attention to pain implicitly, with participants unaware of the relevance or meaning of the pain-related cues.

Notably, the dual-pain CPM paradigm requires participants to deliberately attend to ongoing pain in order to provide continuous verbal ratings. Therefore, explicit attentional tendencies may be more relevant and more directly linked to CPM efficiency measured by self-report than implicit attentional biases. This pattern is consistent with findings from other domains, where implicit and explicit measures influence behavior differently [113,114]. For example, attentional biases toward exercise cues are associated with positive implicit attitudes (i.e., automatic mental associations between an object and its evaluation) toward physical activity. However, only explicit attitudes (i.e., self-reported opinions and intentions) toward physical activity moderate whether the observed attentional biases translate into actual sport behavior [113]. Collectively, these observations raise the hypothesis that explicit attention toward pain may moderate the relationship between attentional biases and CPM efficiency, a possibility that warrants investigation in future studies.

A second difference between our study and Adams et al. (2021) concerns whether attentional biases are assessed while participants are actually experiencing pain. Pain-related cues presented in the absence of pain may be insufficient to capture the complexity of attentional tendencies to pain in healthy individuals, as they do not evoke the motivational significance, emotional arousal, or regulatory demands that characterize real pain [23,26,115]. Assessing attentional biases during ongoing nociceptive input may therefore provide a more sensitive and ecologically valid indication of how individuals naturally prioritize and process pain-relevant information. This point is particularly important given the scarcity of studies that have examined attentional biases in the context of actual pain, despite evidence that such biases are meaningfully related to pain-related cognitions (e.g., fears and catastrophizing) and behaviors [25,116,117]. For instance, Van Ryckeghem et al. (2013) demonstrated that in chronic pain populations, attentional biases interact with daily pain severity to shape disability and distractibility, illustrating that attentional patterns during pain can have meaningful behavioral consequences [117]. This underscores the relevance of further investigating attentional biases under conditions of real nociception to better understand how they manifest and operate in everyday pain experiences [10].

Taken together, our findings emphasize that attention is a multifaceted construct in pain research. Differences between attentional types—such as strategic versus automatic, internal versus external, and explicit versus implicit—likely contribute to inconsistent findings linking attention, pain sensitivity, and pain inhibition. Future studies should measure both implicit and explicit attention, ideally within the same experimental design and in the presence and absence of pain, to clarify how these processes individually and jointly influence endogenous pain modulation in clinical and non-clinical populations.

### 4.4. The impact of emotional distress on HR reactivity and recovery from CPM

In addition to attentional mechanisms, autonomic nervous system (ANS) functioning may provide further insight into the regulatory processes underlying CPM. As expected, HR increased during both tonic heat pain and CPM stimulation and remained elevated during recovery, consistent with prior work on altered autonomic activation during pain modulation [118]. Importantly, our exploratory findings are the first to show how emotional distress influences autonomic reactivity during pain conditioning. Specifically, higher emotional distress was associated with blunted HR increases during CPM. Typically, HR increases during stress reflect adaptive autonomic responses [119], whereas blunted HR reactivity has been linked to difficulties in emotional and pain regulation in conditions such as fibromyalgia, depression, and addiction

[119–121]. For instance, reduced HR reactivity in fibromyalgia predicts greater pain and slower recovery [122]. Together, these findings suggest that psychological distress may impair ANS adaptability, reducing physiological responsiveness to pain and potentially contributing to prolonged emotional recovery following painful experiences [123–125].

Moreover, our results indicate that attentional biases also influence autonomic dynamics, particularly during the recovery phase following CPM. During this post-CPM recovery period, both difficulty disengaging from pain cues and attentional avoidance of pain-related words were associated with elevated HR levels, suggesting delayed autonomic recovery. This pattern may help explain why attentional biases were not associated with CPM magnitude, as their influence appears to emerge primarily during recovery-related processes where affective–physiological regulation plays a major role, instead of during endogenous pain inhibition. Rather than directly impairing the efficacy of CPM, attentional biases may sustain autonomic arousal and hinder recovery, potentially contributing to maladaptive processes such as pain catastrophizing, increased rumination, and heightened negative affect [95,126]. These observations may have important clinical implications, as repeated exposure to pain combined with prolonged autonomic arousal could, over time, exacerbate pain processing through altered expectations and neuroplastic changes in pain-related brain circuits [2,127].

## 5. Limitations and future directions

Several limitations should be noted. Firstly, our sample consisted solely of healthy young women. This limits the generalizability of the findings to males, older individuals, or clinical populations, particularly since age-related HR decline [128], reduced vagal tone [129], and the heightened prevalence of cardiovascular diseases in chronic pain populations [130] could impact pain and autonomic responses. In addition, the relative homogeneity and low psychological distress typically observed in healthy young samples likely restricted variability in both attentional bias indices and CPM responses, reducing the covariance needed to detect associations between them. Secondly, although HR provides a useful index of ANS activity, it only captures a part of the picture, as it is dependent on both parasympathetic and sympathetic activation [118]. Thirdly, psychological models of pain suggest that attentional responses are influenced more by the perceived threat value of pain than by its sensory intensity [88,120]. In addition, trait factors such as anxiety, depression, catastrophizing, and fear of pain can interact with situational factors to shape both attentional and pain responses [10,26,106]. Therefore, future research should examine how attentional biases interact with psychological traits and emotional states and threat perception to affect pain modulation.

Future studies also should incorporate additional physiological markers, such as heart rate variability, skin conductance response, or pupil dilation, to provide a more comprehensive assessment of attentional influences on autonomic regulation in pain modulation [2,118,131]. Besides, clarifying the causal role of attentional biases in recovery is also important. This would help explain their link to autonomic recovery and test whether targeting these biases can improve recovery and reduce long-term pain vulnerability. Notably, in clinical populations, these relationships may differ, as patients often present with deficient CPM inhibition [132,133], heightened attentional biases [23,84,85], or both. Distinguishing such profiles could enable more precise, mechanism-based interventions tailored to each patient's specific pain dysfunction. Finally, replication with standardized attentional tasks that vary cue modality, presentation duration, and perceived threat could help isolate key components of attentional bias. Such work would clarify their role in pain perception and regulation and disentangle the contributions of situational versus dispositional factors.

### 4.4. Conclusion

This study demonstrates that attentional biases to pain are a key predictor of pain sensitivity in healthy females, independent of psychological traits and autonomic responses. While neither attentional biases, psychological traits, nor heart rate dynamics predicted the magnitude of CPM, the observed links between emotional distress, autonomic reactivity, and attentional tendencies point to complex interactions shaping pain regulation and recovery. Taken together, these findings position attentional biases as a central cognitive mechanism in the experience of pain, highlighting their relevance for understanding individual differences in the experience of pain.

## Supporting information

**S1 Text. Supporting Information.** Detailed description of the attentional bias paradigms.
(PDF)

**S1 Table. List of words used in the dot probe task.**
(PDF)

**S2 Table. Mean accuracy and reaction time performance in the attentional bias tasks.**
(PDF)

**S3 Table. Correlations between attentional bias indexes, psychological traits, and emotional state.**
(PDF)

## Acknowledgments

We thank Hagar Bialsky Shmueli, Ayelet Ivtsan, Noa Brave, Farah Samawi, Hagit Kapitanov, Orianne Fridman, Sarit Shaolov, Shir Azulay, and Rani Amit Bar-On for their help on this article. The authors declare no conflicts of interest.

## Author contributions

**Conceptualization:** Einav Gozansky, Irit Weissman-Fogel.

**Data curation:** Einav Gozansky.

**Formal analysis:** Einav Gozansky, Irit Weissman-Fogel, Hadas Okon-Singer.

**Funding acquisition:** Einav Gozansky, Irit Weissman-Fogel, Hadas Okon-Singer.

**Investigation:** Hadas Okon-Singer.

**Methodology:** Einav Gozansky, Irit Weissman-Fogel, Hadas Okon-Singer.

**Project administration:** Einav Gozansky, Irit Weissman-Fogel, Hadas Okon-Singer.

**Resources:** Einav Gozansky.

**Software:** Einav Gozansky.

**Supervision:** Irit Weissman-Fogel, Hadas Okon-Singer.

**Validation:** Einav Gozansky, Irit Weissman-Fogel, Hadas Okon-Singer.

**Visualization:** Einav Gozansky.

**Writing – original draft:** Einav Gozansky.

**Writing – review & editing:** Einav Gozansky, Irit Weissman-Fogel, Hadas Okon-Singer.

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
