## [Decision Letter · Decision Letter 0]

18 Nov 2025

Dear Dr. Gozansky,

Thank you for submitting your manuscript to PLOS ONE. After careful consideration, we feel that it has merit but does not fully meet PLOS ONE’s publication criteria as it currently stands. Therefore, we invite you to submit a revised version of the manuscript that addresses the points raised during the review process.

We look forward to receiving your revised manuscript.

Kind regards,

Xianwei Che

Academic Editor

PLOS ONE

Journal Requirements:

3. Please note that your Data Availability Statement is currently missing the DOI. If your manuscript is accepted for publication, you will be asked to provide these details on a very short timeline. We therefore suggest that you provide this information now, though we will not hold up the peer review process if you are unable.

Reviewers' comments:

Reviewer's Responses to Questions

**Comments to the Author**

1. Is the manuscript technically sound, and do the data support the conclusions?

Reviewer #1: Yes

Reviewer #2: Yes

Reviewer #3: Yes

2. Has the statistical analysis been performed appropriately and rigorously?

Reviewer #1: Yes

Reviewer #2: Yes

Reviewer #3: Yes

3. Have the authors made all data underlying the findings in their manuscript fully available?

Reviewer #1: Yes

Reviewer #2: Yes

Reviewer #3: Yes

4. Is the manuscript presented in an intelligible fashion and written in standard English?

Reviewer #1: Yes

Reviewer #2: Yes

Reviewer #3: Yes

Reviewer #1: The authors investigated the interplay between pain-related attentional biases, psychological traits, heart rate as a measure of autonomic reactivity and conditioned pain modulation (CPM). The relevance of the current study becomes clear in the context of previous research; the authors did a great job in summarizing the state of the art and identifying important gaps in knowledge which they attempted to close.

From my perspective, this study has several important strengths that I would like to point out before sharing my criticism:

- The authors used two different paradigms for assessing attentional bias combined with two different sets of stimuli (pain-related words vs. pictures). This approach is warranted by previous research and led to the identification of a very interesting pattern in results which would have been missed when using just one of the two paradigms.

- The inclusion of heart rate in their analyses yielded an innovative result linking attentional bias and psychological distress to autonomic reactivity under CPM/painful stimulation which is highly relevant and stimulates further investigation of these associations.

- The overall writing style is clear and concise, the theoretical and empirical background is presented extensively, and the statistical analyses are thorough and suitable for testing the authors' hypotheses.

However, I have a few comments that I would like the authors to address:

1) When the authors state their hypothesis at the end of the Introduction, the hierarchy of and potential interaction between the different predictors does not become entirely clear (‘We hypothesized that individuals with stronger attentional biases toward pain, higher levels of pain-related distress, and heightened autonomic reactivity during CPM would exhibit reduced endogenous pain inhibition’). In contrast, in the first paragraph of the Discussion it is clearly stated that attentional bias is treated as the main predictor, and it was explored whether the other factors provide ‘additional explanatory value’. I would recommend adjusting the wording in the Introduction accordingly to make this clearer. This might also apply to the Abstract.

2) I was confused by the following sentence: ‘Specifically, the ANS is involved in shaping CPM efficacy, with studies showing increased heart rate (HR) and sympathetic activation during CPM’ (Introduction, p. 5) as the first part of the sentence does not logically follow from the second part. Did these studies only show that heart rate was elevated during CPM, or are there clear indicators for an involvement in CPM efficacy (e.g., participants with higher heart rate showing more efficient CPM)? It would be important to clarify this.

3) My main concerns relate to the paradigm used for CPM/painful stimulation. Here, I noticed a few aspects that are (from my experience) unusual and might be problematic:

a) Individually adjusting the intensity of the test stimuli to a level of 60 seems reasonable; however, I am not sure about the reliability of the authors’ methods. In my experience, it takes quite some time to familiarize participants with the thermode and several practice runs so they learn to differentiate between non-painful and painful sensation and to reliably use the rating scale. Applying three single stimuli to determine the approximate location of a ‘60’ on the rating scale (then followed by three more) seems insufficient. Unfortunately, the ratings of the test stimuli (Table S 2 in the Supplementary Material) underscore this problem, with ratings decreasing dramatically from 60 to 20-25 within the 60 seconds of stimulation. In my eyes, this suggests that the initial rating might be exaggerated by fear/startle. The authors should at least discuss this and the descriptive statistics of the test stimulus ratings should be presented in the Results section (not in the Supplementary Material) as they are the main outcome measure of this study.

b) Relating to this, please include the stimulus duration of the stimuli that were used to determine ‘pain 60’, the verbal anchors (if any) of the pain scale and the resulting stimulation intensities for ‘pain 60’.

c) I am not sure why the authors chose to individually tailor the temperature of the hot water bath (conditioning stimulus) as most studies use a fixed temperature of 46 or 46.4 °C. However, please provide some information on the procedure used. How long was the hand immersed during these trials? Could this have influenced the CPM response by potential adaptation to the conditioning stimulus? In addition, the descriptive ratings of CS ratings should also be provided in the Results section.

d) Why were participants explicitly instructed to focus their attention on the test stimulus? This also seems unusual and potentially problematic as the frequent ratings of the TS (every 10 s) probably primed attention to the TS anyway. The authors state that instructions to focus on the CS can increase CPM; if this works the other way round as well, it might have led to a suppression of CPM effects.

e) It is a bit unfortunate that the authors failed to obtain a classical CPM effect (main effect of condition) which might be due to the factors mentioned above. However, the time-dependent effect seems interesting (adaptation over time in both conditions, but stronger in the CPM condition) – especially in the light of previous research showing that CPM can abolish temporal summation – and could maybe be discussed a bit more.

4) On p. 12, it is stated that ‘the THP test stimulus was applied for 60 seconds to the dominant hand’; I suppose the authors mean ‘the forearm of the dominant hand’ as described in the preceding paragraph?

5) At some points, I was a bit confused by the terminology used for describing the experimental conditions. Personally, I prefer the terminology ‘baseline condition’ (for TS alone) and ‘CPM condition’ (for TS + CS), but I see how that would be a problem here as ‘baseline’ is used to refer to the initial HR assessment. However, I think that the term ‘THP’ could be confusing because it seems unclear if it refers to a stimulus type or a condition; this caused some confusion for me when looking at the descriptives given in Table S2. I also had to scroll back and forth a few times to understand the baseline/reactivity/recovery phases. Maybe the terminology and/or Figure 1 could be improved to simplify and clarify this.

Reviewer #2: Summary

This study (N=86 healthy women) tests whether attentional biases (Perceptual Load images, Dot-Probe words), psychological traits (DASS-21, PCS, FPQ), and autonomic activity (heart rate; baseline/reactivity/recovery) explain individual differences in pain sensitivity and conditioned pain modulation (CPM). Key findings: stronger interference by pain cues under high load → higher pain sensitivity; Dot-Probe avoidance → higher fear of pain; none of the examined factors predicted CPM magnitude. HR increased during CPM and stayed elevated in recovery; higher distress related to blunted HR reactivity; attentional indices predicted higher HR during recovery.

Strengths

• Clear gap and two-task design. Nicely motivates moving beyond attentional focus to biases, using complementary tasks that target earlier capture vs strategic avoidance.

• Well-specified QST/CPM and ECG stages. Baseline/reactivity/recovery windows and repeated-measures analyses are clear; HR effects are quantified.

• Integration of psychology and ANS. Parallel treatment of distress, fear, catastrophizing with HR dynamics strengthens interpretation.

Areas for Improvement:

1. The fact that attentional biases wasn’t correlated with CPM is unexpected. I would suggest authors to make a more scholarly discussion of this. Which could include the fact that task/modality/time differences could dilute shared variance; attentional tendencies may relate more to autonomic recovery than to CPM magnitude, which is consistent with your recovery HR findings; and most importantly that restricted range in healthy young women reduces covariance.

2. The authors have done an exemplary job providing comprehensive methodological details, which greatly enhances transparency and reproducibility. However, the level of detail in the main text occasionally impedes narrative flow and may overwhelm readers unfamiliar with attentional bias paradigms or QST protocols. I recommend moving some procedural specifics to supplementary materials while retaining core conceptual information in the main text. For example, Section 2.3.1.a (Perceptual Load task, lines 156-182) could be streamlined to emphasize what the task measures (early attentional capture by pain stimuli under varying cognitive load) and why it was chosen, with stimulus presentation parameters, letter selection criteria, and catch trial percentages moved to a supplement. The same could be done for other procedures. This restructuring would help readers grasp the conceptual logic—how attentional biases were operationalized and measured—without getting lost in technical specifications, while still providing the detail necessary for replication.

3. While the authors report η² for their ANOVA analyses, I recommend also including Cohen's d effect sizes for key pairwise comparisons and t-tests to facilitate meta-analytic integration and practical interpretation. It is recommended to report both η² for omnibus tests and Cohen's d (or Hedges' g for small samples) for specific contrasts, as this provides readers with both variance explained and the magnitude of differences in standard deviation units. This is particularly important given that some findings (e.g., attentional biases predicting pain sensitivity) represent novel contributions that future researchers will want to interpret and replicate.

4. The sentence “Specifically, the ANS is involved in shaping CPM efficacy, with studies showing increased 96 heart rate (HR) and sympathetic activation during CPM: should be revised to distinguish group-level HR increases “during CPM” from between-person prediction. Do authors mean that the expected within-task HR rise or that individual HR reactivity predicts CPM magnitude.

5. At first mention in Participants, add a parenthetical brief explanation regarding SONA, clarifying its meaning. This makes recruitment transparent for international readers and non-academic audiences.

6. Authors describe that normality was checked using Shapiro-Wilk and histograms but don’t describe what was done if violated.

Minor Comments:

1. The resolution (DPI) of the figures appears to be below publication standards. Panel labels in Figure 6 (A, B, C) are difficult to read, and the violin plot contours show pixelation.

2. The OSF data repository link provided (https://osf.io/3bwy7/files/osfstorage) does not appear to be functioning properly or may not be publicly accessible. When accessing the link, it does not resolve to a viewable repository page. I believe the correct link should be: https://osf.io/3bwy7/files or https://osf.io/3bwy7.

This manuscript represents a methodologically rigorous and theoretically well-motivated investigation into the role of attentional biases in pain sensitivity and endogenous pain modulation. The study makes meaningful contributions by demonstrating that attentional biases predict pain sensitivity independently of psychological factors, and by revealing their association with autonomic recovery dynamics, novel findings that advance our understanding of cognitive mechanisms in pain processing. The use of complementary attentional bias paradigms, careful QST/CPM methodology, and integration of autonomic measures are particular strengths. The suggested revisions focus primarily on enhancing clarity in the Discussion of unexpected null results, improving readability through strategic reorganization of methodological detail, strengthening effect size reporting, and addressing minor technical issues with figures and data accessibility.

Reviewer #3: This study investigated whether attentional bias predicts individual differences in pain sensitivity and endogenous pain modulation, and whether psychological, emotional, and autonomic factors provide additional explanatory value. The authors are to be congratulated for this work. The methods in particular were very clearly described. Well done!

I have only a few minor queries:

- The quality of the figures seems to be low; they are quite pixelated. I recommend that you increase the resolution of the images.

- Section 2.4.1: please provide a justification for including only participants with 2.5 z-scores above or below each participant’s average RT. The rationale for this approach is not entirely clear.

- The relevance of including BMI and ethnicity are unclear. Please can you include a justification for the relevance of these outcomes.

- In Table 1, it would be useful for the reader if you include the possible ranges of the questionnaires so that the reader can better interpret your summary statistics.

- Please can you include assessments of model fitness for your regression analyses.

- Was the protocol/statistical analysis plan locked online prior to conducting the analysis? In acknowledgement of the principles of open science, I'd recommend reporting on this.

**Do you want your identity to be public for this peer review?** For information about this choice, including consent withdrawal, please see our Privacy Policy

Reviewer #1: No

Reviewer #2: No

Reviewer #3: No

---

## [Author Response · Author response to Decision Letter 1]

27 Dec 2025

Response letter to the reviewers’ comments

Manuscript ID: PONE-D-25-45053

The involvement of attentional biases in endogenous pain inhibition and autonomic reactivity

Dear Prof. Xianwei Che, Academic Editor, and Reviewers,

We sincerely thank the reviewers for their thoughtful, constructive, and highly encouraging feedback on our manuscript. We truly appreciate the kind words and the evident enthusiasm and engagement shown in reading our work. The comments provided were insightful and have significantly strengthened the quality and clarity of the manuscript.

We have addressed all concerns raised by the reviewers in a point-by-point manner below. All revisions are marked using highlights in the updated manuscript (and cited within the corresponding responses for clarity).

Reviewer #1

“The authors investigated the interplay between pain-related attentional biases, psychological traits, heart rate as a measure of autonomic reactivity and conditioned pain modulation (CPM). The relevance of the current study becomes clear in the context of previous research; the authors did a great job in summarizing the state of the art and identifying important gaps in knowledge which they attempted to close.

From my perspective, this study has several important strengths that I would like to point out before sharing my criticism:

- The authors used two different paradigms for assessing attentional bias combined with two different sets of stimuli (pain-related words vs. pictures). This approach is warranted by previous research and led to the identification of a very interesting pattern in results, which would have been missed when using just one of the two paradigms.

- The inclusion of heart rate in their analyses yielded an innovative result linking attentional bias and psychological distress to autonomic reactivity under CPM/painful stimulation which is highly relevant and stimulates further investigation of these associations.

- The overall writing style is clear and concise, the theoretical and empirical background is presented extensively, and the statistical analyses are thorough and suitable for testing the authors' hypotheses.”

Comment 1: “When the authors state their hypothesis at the end of the Introduction, the hierarchy of and potential interaction between the different predictors does not become entirely clear (‘We hypothesized that individuals with stronger attentional biases toward pain, higher levels of pain-related distress, and heightened autonomic reactivity during CPM would exhibit reduced endogenous pain inhibition’). In contrast, in the first paragraph of the Discussion it is clearly stated that attentional bias is treated as the main predictor, and it was explored whether the other factors provide ‘additional explanatory value’. I would recommend adjusting the wording in the Introduction accordingly to make this clearer. This might also apply to the Abstract.”

Response: We thank the reviewer for this important and constructive comment. We agree that clarifying the hierarchy of predictors in the introduction and abstract will improve the manuscript. Accordingly, we have revised the wording of the study aims and hypothesis to clearly indicate that attentional bias is treated as the primary predictor, while psychological, emotional, and autonomic factors are examined for their additional explanatory value. These revisions now align more closely with the summary lines presented in the discussion.

Introduction, Page 6, paragraph 3: ”This study aimed to determine whether attentional bias accounts for individual differences in pain sensitivity and endogenous pain modulation, and to what extent psychological, emotional, and autonomic factors offer additional explanatory power. We focused particularly on CPM as an index of endogenous pain inhibition and HR dynamics as a marker of autonomic activity. We primarily hypothesized that individuals with stronger attentional biases toward pain would exhibit reduced endogenous pain inhibition. In addition, we expected that higher levels of pain-related distress, heightened autonomic reactivity during CPM, would contribute secondary, complementary effects, further explaining individual variability in endogenous pain modulation.”

Abstract, Page 2: “… The present study examined whether attentional biases predict individual differences in pain sensitivity and endogenous pain modulation, and whether psychological, emotional, and autonomic factors are associated with these outcomes.”

Comment 2: “I was confused by the following sentence: ‘Specifically, the ANS is involved in shaping CPM efficacy, with studies showing increased heart rate (HR) and sympathetic activation during CPM’ (Introduction, p. 5) as the first part of the sentence does not logically follow from the second part. Did these studies only show that heart rate was elevated during CPM, or are there clear indicators for an involvement in CPM efficacy (e.g., participants with higher heart rate showing more efficient CPM)? It would be important to clarify this.”

Response: We thank the reviewer for highlighting this ambiguity. We revised the section to clearly distinguish evidence for autonomic nervous system (ANS) predictors of CPM efficacy from autonomic reactivity during the CPM task itself. The revised text now notes that several studies have linked resting vagal tone and sympathetic reactivity during test pain alone to CPM magnitude, supporting a predictive role of ANS activity in CPM efficacy. In contrast, only a small number of studies have assessed autonomic changes during the CPM paradigm, reporting increased sympathetic activation and reduced vagal tone at the group level. Importantly, only one of these studies examined the association between autonomic reactivity during CPM and CPM magnitude and found no significant relationship. This revision resolves the inconsistency and clarifies the distinction between predictive and within-task autonomic indicators.

Page 5, Paragraph 3: “In addition, autonomic nervous system (ANS) activity is increasingly recognized as a key contributor to the subjective experience of pain [19,99] and pain inhibition [13,53,63]. Specifically, initial evidence suggests that the ANS may be involved in shaping CPM efficacy. A few studies have linked resting vagal tone, measured via heart rate variability (HRV) or blood pressure, to later CPM magnitude, with higher vagal tone predicting a larger CPM effect [69,72,76]. Other work highlights the role of sympathetic activation, showing that greater sympathetic reactivity during pain alone is associated with enhanced CPM efficacy [15,119]. At the same time, cardiovascular autonomic dynamics during CPM have been investigated in only two studies to date [5,51]. Both reported group-level decreases in vagal tone and increases in sympathetic activation during the CPM procedure; however, only one study examined whether such changes were associated with CPM magnitude, and it found no significant correlation [51]. “

Comment 3.a: “My main concerns relate to the paradigm used for CPM/painful stimulation. Here, I noticed a few aspects that are (from my experience) unusual and might be problematic:

a) Individually adjusting the intensity of the test stimuli to a level of 60 seems reasonable; however, I am not sure about the reliability of the authors’ methods. In my experience, it takes quite some time to familiarize participants with the thermode and several practice runs so they learn to differentiate between non-painful and painful sensation and to reliably use the rating scale. Applying three single stimuli to determine the approximate location of a ‘60’ on the rating scale (then followed by three more) seems insufficient. Unfortunately, the ratings of the test stimuli (Table S 2 in the Supplementary Material) underscore this problem, with ratings decreasing dramatically from 60 to 20-25 within the 60 seconds of stimulation. In my eyes, this suggests that the initial rating might be exaggerated by fear/startle. The authors should at least discuss this and the descriptive statistics of the test stimulus ratings should be presented in the Results section (not in the Supplementary Material) as they are the main outcome measure of this study”.

Response: The reviewer raises an important methodological consideration, highlighting the need to clarify our procedure for determining the pain-60 stimulus. We agree that the original description may have given the impression that only a limited number of trials were used, which could raise concerns regarding reliability and participant familiarization.

In practice, the calibration procedure was lengthy and iterative, typically lasting approximately 15 minutes. Participants were exposed to repeated blocks of three heat stimuli at varying temperatures, and temperature adjustments were made between blocks based on the participants’ numerical pain scale (NPS) ratings, following commonly used procedures (e.g., Granot et al., 2008; https://doi.org/10.1016/j.pain.2007.06.029). This process was repeated until the same temperature elicited an NPS rating of approximately 60 in three separate repetitions, at which point it was accepted as the participant’s pain-60 level. This approach ensured adequate familiarization with both the thermode and the rating scale, as well as stability of the selected stimulus intensity. We have revised the Methods section to describe this procedure more explicitly.

It should also be noted that although a fixed stimulus intensity is determined based on pain-60—reflecting the perceived pain intensity at the onset of stimulation—pain ratings during prolonged tonic heat stimulation are expected to vary over time as a function of stimulus duration and stimulation sequence. In our sample, ratings declined on average from the initial target value (~60) toward lower levels (approximately 20–25) over the 60-second stimulation period, with substantial interindividual variability. Such temporal changes in pain ratings are well documented in prolonged heat pain paradigms and reflect known dynamics of pain perception, including adaptation or sensitization depending on stimulus temperature and individual pain sensitivity (e.g., Weissman-Fogel et al., 2015; https://doi.org/10.1002/ejp.562), rather than exaggerated initial ratings due to fear or startle alone. Importantly, higher pain-60 temperatures are typically associated with greater sensitization, whereas lower pain-60 temperatures tend to be associated with adaptation. However, this does not compromise the validity of the CPM measure, as CPM is computed using a subtraction-based index that inherently accounts for baseline changes and adaptation effects.

Furthermore, in response to the reviewer’s suggestion, we have moved the descriptive statistics of the test-stimulus ratings from the Supplementary Material to the Results section, Table 2.

Page 11, Paragraph 1: “A 30 × 30 cm Peltier thermode (Thermal Sensory Analyzer-2001, Medoc, Israel) was applied to the volar surface of the dominant forearm, to determine the heat temperature corresponding to pain-60 (i.e., the temperature participants rate as 60 on a 0-100 numerical pain scale (NPS), ranging from '0'=no pain, '100'=worst imaginable pain). Stimulus intensity was individually calibrated such that all participants experienced the same subjective level of pain, as the focus of the study was on pain perception rather than nociceptive input. The pain-60 calibration procedure employed here is consistent with the commonly used protocol in the literature [42]. In general, participants were exposed to a series of hot stimuli of 8-second duration. Participants were first familiarized with the NPS by rating three initial heat stimuli (43 °C, 45 °C, and 47 °C; ramp rate: 2 °C/s from a 32 °C baseline), with a 20-seconds inter-stimulus interval. After each stimulus, participants were asked to report their level of pain on the verbal NPS. Based on these ratings, we identified an initial temperature with the NPS score closest to 60, with which we continued our pain-60 determination procedure.

The pain-60 determination then continued with a series of blocks containing three heat stimuli: the closest temperature to the temperature rated as 60 at the introduction phase, one degree lower, and one degree higher (presented in a random order). After each trial, the thermode was repositioned to prevent sensitization. At the end of each rating block, if none of the three temperatures elicited a rating near 60, another block containing three ratings was conducted, with the heat stimuli temperatures adjusted upward or downward in 1 °C increments, again including the adjacent temperatures above and below the new center point.

Importantly, this procedure was iterative: blocks were repeatedly administered and adjusted as needed until the same temperature consistently produced an NPS rating of approximately 60 in three separate trials. This full calibration procedure typically requires around 15 minutes to be completed. Only after achieving such consistency was the temperature defined as the participant’s pain-60, and it was used for the THP stimulation applied alone (“test-stimulus alone”) and during CPM (“test-stimulus under conditioning stimulus”, see below for further details). It should be noted that although a fixed stimulus intensity is used based on the determination of pain-60, which reflects the perceived pain intensity at the onset of stimulation, reported pain ratings typically vary over the course of the test sessions as a function of several factors, including stimulus duration and stimulation sequence [72,130]. Notably, higher pain-60 temperatures are associated with greater sensitization, whereas lower pain-60 temperatures tend to be associated with adaptation; however, this does not compromise the validity of the CPM measure, as CPM is calculated using a subtraction-based index that inherently accounts for adaptation effects.”

Comment 3.b.: “Relating to this, please include the stimulus duration of the stimuli that were used to determine ‘pain 60’, the verbal anchors (if any) of the pain scale, and the resulting stimulation intensities for ‘pain 60”.

Response: Thank you for the clarifying request. The stimulus duration, the verbal anchors of the NPS, and the resulting temperatures used for determining pain-60 are important methodological details and were unintentionally omitted in the initial submission. We have now added this information clearly in the Methods section and have also included the descriptive statistics for the resulting pain-60 temperatures in the Results section, as suggested.

Page 11, Paragraph 1: …” to determine the heat temperature corresponding to pain-60 (i.e., the temperature participants rate as 60 on a 0-100 numerical pain scale (NPS), ranging from '0'=no pain, '100'=worst imaginable pain). Participants were first familiarized with the NPS by rating three initial 8-second heat stimuli (43 °C, 45 °C, and 47 °C; ramp rate: 2 °C/s from a 32 °C baseline). “

Page 20, Paragraph 3: “The pain-60 temperatures used for the THP test ranged between 41-48°C with a mean temperature of 45.04°C (1.95).”

Comment 3.c.: “I am not sure why the authors chose to individually tailor the temperature of the hot water bath (conditioning stimulus) as most studies use a fixed temperature of 46 or 46.4 °C. However, please provide some information on the procedure used. How long was the hand immersed during these trials? Could this have influenced the CPM response by potential adaptation to the conditioning stimulus? In addition, the descriptive ratings of CS ratings should also be provided in the Results section.”

Response: We would like to address the reviewer’s concern regarding the individual tailoring of the conditioning stimuli temperature. At the start of data collection, we observed substantial inter-individual variability in pain responses to the fixed temperature of 45.5 °C: for some participants, this temperature was too intense to maintain hand immersion, whereas for others it produced little to no pain. For this reason, based on Nir et al. (2011; https://doi.org/10.1016/j.ejpain.2010.10.001), indicating the CPM effect is observed only when hot water immersion leads to moderate or intense pain, we adopted an indi

---

## [Decision Letter · Decision Letter 1]

19 Jan 2026

The involvement of attentional biases in endogenous pain inhibition and autonomic reactivity

PONE-D-25-45053R1

Dear Dr. Gozansky,

We’re pleased to inform you that your manuscript has been judged scientifically suitable for publication and will be formally accepted for publication once it meets all outstanding technical requirements.

Kind regards,

Xianwei Che

Academic Editor

PLOS One

Additional Editor Comments (optional):

Reviewers' comments:

Reviewer's Responses to Questions

**Comments to the Author**

Reviewer #1: All comments have been addressed

Reviewer #3: All comments have been addressed

2. Is the manuscript technically sound, and do the data support the conclusions?

Reviewer #1: Yes

Reviewer #3: Yes

3. Has the statistical analysis been performed appropriately and rigorously?

Reviewer #1: Yes

Reviewer #3: Yes

4. Have the authors made all data underlying the findings in their manuscript fully available?

Reviewer #1: Yes

Reviewer #3: Yes

5. Is the manuscript presented in an intelligible fashion and written in standard English?

Reviewer #1: Yes

Reviewer #3: Yes

Reviewer #1: The authors have addressed all of my concerns and greatly improved the manuscript. I have no further comments

Reviewer #3: (No Response)

**Do you want your identity to be public for this peer review?** For information about this choice, including consent withdrawal, please see our Privacy Policy

Reviewer #1: No

Reviewer #3: No

---

## [Editor Report · Acceptance letter]

PONE-D-25-45053R1

PLOS One

Dear Dr. Gozansky,

I'm pleased to inform you that your manuscript has been deemed suitable for publication in PLOS One. Congratulations! Your manuscript is now being handed over to our production team.

Kind regards,

on behalf of

Professor Xianwei Che

Academic Editor

PLOS One